# Pharmaceutical Approach to Develop Novel Photosensitizer Nanoformulation: An Example of Design and Characterization Rationale of Chlorophyll α Derivative

**DOI:** 10.3390/pharmaceutics16010126

**Published:** 2024-01-18

**Authors:** Maria B. Sokol, Veronika A. Beganovskaya, Mariia R. Mollaeva, Nikita G. Yabbarov, Margarita V. Chirkina, Dmitry V. Belykh, Olga M. Startseva, Anton E. Egorov, Alexey A. Kostyukov, Vladimir A. Kuzmin, Sergei M. Lomakin, Natalia G. Shilkina, Alexey V. Krivandin, Olga V. Shatalova, Margarita A. Gradova, Maxim A. Abakumov, Aleksey A. Nikitin, Varvara P. Maksimova, Kirill I. Kirsanov, Elena D. Nikolskaya

**Affiliations:** 1N. M. Emanuel Institute of Biochemical Physics of Russian Academy of Sciences, 119334 Moscow, Russia; mariyabsokol@gmail.com (M.B.S.); veronika.beganovckaya@bk.ru (V.A.B.); mollaevamariia@gmail.com (M.R.M.); chir.marg@mail.ru (M.V.C.); ae.yegorov@gmail.com (A.E.E.); vladimirkuzmin7@gmail.com (V.A.K.); lomakin@sky.chph.ras.ru (S.M.L.); a.krivandin@sky.chph.ras.ru (A.V.K.); shatalova@sky.chph.ras.ru (O.V.S.); 2Institute of Chemistry, Komi Scientific Center, Ural Division of the Russian Academy of Sciences, 167982 Syktyvkar, Russia; belykh-dv@mail.ru; 3Pitirim Sorokin Syktyvkar State University, 167001 Syktyvkar, Russia; om_startseva@mail.ru; 4National Research Nuclear University MEPhI, 115409 Moscow, Russia; 5N. N. Semenov Federal Research Center for Chemical Physics of Russian Academy of Sciences, 119991 Moscow, Russia; tashi05@list.ru (N.G.S.);; 6Laboratory of Biomedical Nanomaterials, National University of Science and Technology (MISIS), 119049 Moscow, Russia; abakumov.ma@misis.ru (M.A.A.); nikitin.aa@misis.ru (A.A.N.); 7Blokhin National Medical Research Center of Oncology, 115478 Moscow, Russia; lavarvar@gmail.com (V.P.M.); kkirsanov85@yandex.ru (K.I.K.)

**Keywords:** photodynamic therapy, photosensitizer, nanoparticles, PLGA, chlorophyll derivative, antitumor activity, cell death

## Abstract

In this study, we described physico-chemical properties of novel nanoformulation of photosensitizer-pyropheophorbide α 17-diethylene glycol ester (XL) (chlorophyll α derivative), revealing insights into antitumor activity and maintaining quality, meeting the pharmaceutical approach of new nanoformulation design. Our formulation, based on poly(lactic-co-glycolic acid) (PLGA) nanoparticles, increased XL solubility and selective tumor-targeted accumulation. In our research, we revealed, for the first time, that XL binding to polyvinyl alcohol (PVA) enhances XL photophysical activity, providing the rationale for PVA application as a stabilizer for nanoformulations. Results of FTIR, DSC, and XRD revealed the physical interactions between XL and excipients, including PVA, indicating that the encapsulation maintained XL binding to PVA. The encapsulated XL exhibited higher photophysical activity compared to non-encapsulated substance, which can be attributed to the influence of residual PVA. Gamma-irradiation led to degradation of XL; however, successful sterilization of the samples was achieved through the filtration. Importantly, the encapsulated and sterilized XL retained cytotoxicity against both 2D and 3D tumor cell models, demonstrating the potential of the formulated NP–XL for photodynamic therapy applications, but lacked the ability to reactivate epigenetically silenced genes. These findings provide valuable insights into the design and characterization of PLGA-based nanoparticles for the encapsulation of photosensitizers.

## 1. Introduction

Photodynamic therapy (PDT) has emerged as a highly sought-after oncology treatment, with earliest applications dating back to the early 1900s [1]. PDT effectively targets superficial neoplasms and microscopic residual tumors after resection, making it a valuable therapeutic modality [2,3]. This treatment modality involves the local activation of a photosensitizer (PS) by visible red light, triggering the production of reactive oxygen species that selectively kill cancer cells. Additionally, PDT can disrupt tumor blood vessels, impeding tumor growth. For optimal PDT performance, an ideal PS should possess specific characteristics: (1) selective accumulation in neoplasm tissues, (2) intense absorption in the red or near-infrared region of the spectrum to maximize tissue penetration of light, (3) efficient generation of reactive oxygen species without aggregation in aqueous solutions, (4) minimal general toxicity, and (5) intense fluorescence for simultaneous fluorescence diagnostics.

In current clinical practice, second-generation PS such as chlorins, chlorophylls, and chlorophylls-like sensitizers are widely applied due to their absorption maximum in the far-red region and effective singlet molecular oxygen production [4].

However, recent research efforts have focused on novel PS with an extra feature design—a good hydrophilic–lipophilic balance. The ideal PS exhibits amphiphilic properties, demonstrating both water- and lipid-solubility, possess low dark and pronounced photocytotoxicity, stablility during storage, and cost-effectiveness [1].

In line with these trends, Belykh and co-authors reported the synthesis of a novel chlorophyll α derivative called pyropheophorbide α 17-diethylene glycol ester (XL), which combines a hydrophobic macrocycle with a hydrophilic oligoethylene glycol substituent (Figure 1a) [5]. Consisting of the hydrophobic moiety and hydrophilic substituent, the structure of XL exhibited amphiphilic properties, which enable integration into various supramolecular assemblies. In vitro experiments showed that XL displayed high photoactivity, inducing reactive oxygen species formation, mitochondrial depolarization, and apoptosis [6]. These findings highlight the promising potential of XL in the field of PDT.

Localization near or within tumor cells is essential for PS to exert optimal photodynamic activity. Despite macrocycle modification, XL still exhibits water solubility limitations, reducing bioavailability. To address these challenges, we enhanced the selective accumulation of XL in cancer cells and improved bioavailability by design of XL-loaded nanoparticles using poly(lactic-co-glycolic acid) (PLGA). Among numerous materials available for modern nanodrug design, we selected PLGA, because the polymer is approved for medical application and applied widely in surgery, orthopedy, traumatology, and drug delivery [7]. Notably, PLGA nanoparticles have a propensity to accumulate in tumor tissues through the enhanced permeability and retention (EPR) effect, leveraging the altered vasculature and compromised blood flow in tumors [8]. This enables specific extravasation and intracellular delivery of PLGA nanoparticles and augments the anticancer potential of encapsulated PS in PDT.

Previously, we performed several studies describing formulation and biological effects evaluation of several perspective reactive oxygen species (ROS) modulators based on pheophorbide [6], tetraphenyl-porphyrin metal complexes [9,10,11], and their polymer entrapped forms. The main problems of these classes of molecules are high lipophilicity, which limits effective application and potential therapeutic effect realization in vitro and in vivo [12]. We demonstrated efficacy of PLGA-entrapment approach, which allowed us to enhance solubility significantly in physiological media; design, develop and validate complex approaches to PLGA-nanoparticles containing Me-porphyrins formulation process [9]; stabilize and properly evaluate physico-chemical properties under physiological conditions; and perform reproducible and safe analysis of in vitro and in vivo activity of Me-porphyrins [9,10,11,13], XL [6], and bacteriopurpurinimide [14]. PLGA nanoparticles containing Fe^III^Cl-tetraphenylporphyrin in the presence of ascorbic acid (AA), representing binary catalyst therapy (BCT), system effectively impaired ROS balance in MCF-7, K-562, and MG-63 cancer cells leading to apoptosis in vitro; in vivo, the combination was safe and highly effective against murine P-388 tumor model [11,13]. BCT systems based on Mn^III^-tetraphenylporphyrin chloride-loaded PLGA nanoparticles and AA endocytosed by 4T1, SKOV-3, K-562, and HeLa cancer cells in vitro effectively generate intracellular H_2_O_2_ and leading to cell death via caspase 3/9 activation and AIF translocation to nuclei [10]. The photosensitizers based on XL or bacteriopurpurinimide PLGA-loaded nanoparticles revealed high-level photoinduced cytotoxicity in vitro via initiation of DNA damage, glutathione depletion, and excessive lipid peroxidation, and were effective in vivo against HeLa and S-37 tumor models, respectively [6,14].

Herein, we reported for the first time the main physicochemical properties of XL–NP to (1) provide a rationale on excipient selection and (2) get a complete understanding of XL–NP properties for further prediction of the formulation behavior in the body. As highlighted in different aspects of quality by design [14], these points are crucial in the nanopharmaceutical approach since they provide a first glimpse on storage stability, sterilization methods for parenteral administration, drug dosing, and pharmacokinetics. Thus, in our study, we focused on XL–NP further practical application. Listed tests and comprehensive descriptions can be used as specification for nanoformulation quality assurance. This approach would be of interest for similar nanoformulation of photosensitizers.

In this article, we performed a detailed investigation of XL photophysical properties both before and after encapsulation; we described a rationale on emulsion stabilizer selection for XL-loaded PLGA nanoparticles and sterilization technique. We also confirmed our previous data of XL–NP dark and light cytotoxicity and performed further analysis by applying a complex model. We applied 3D models of human triple-negative breast cancer cells MDA-MB-231. Another model we used was an epigenetically active cell line. Epigenetic modulators from the class of histone deacetylase inhibitors (HDACs) and DNA methyltransferase inhibitors (DNMTs) are currently approved for anticancer therapy. Epigenetic remodeling combined with PDT represent effective mechanisms leading to development of anticancer immunity and potentiated anticancer effects. Compounds with both photodynamic properties and epigenetic activity are of great interest for antitumor therapy. Previously, no such drugs have been reported. Thus, we investigated the epigenetic activity of XL and XL-loaded PLGA nanoparticles using HeLa TI cell-based assay. Our study provides a better understanding of underlying phototoxic XL–NP properties.

## 2. Materials and Methods

### 2.1. Materials

Pyropheophorbide α 17-diethylene glycol ester (XL) was synthesized as previously described [5]. Acid terminated poly(D,L-lactide-co-glycolide) with a monomer ratio of 50:50, MW 17,000–21,000, and inherent viscosity of 0.2 dL/g was purchased from LACTEL Absorbance Polymers (Birmingham, AL, USA). D-mannitol, polyvinyl alcohol (PVA, MW 30,000–70,000, 87–90% hydrolyzed) and trifluoroacetic acid were purchased from Sigma-Aldrich (St. Louis, MO, USA). Chloroform, dichloromethane, boric acid, ethanol, HCl, and NaOH were purchased from Ruskhim (Moscow, Russia). Acetonitrile was obtained from Fisher Scientific (Loughborough, UK). Dimethyl sulphoxide (DMSO), Tween 80, and phosphate-buffered saline (PBS) were purchased from Amreso (Solon, OH, USA). Triton X100 and sodium dodecyl sulfate (SDS) were purchased from Serva (Heidelberg, Germany). Heparin was obtained from Khimmed Syntes (Moscow, Russia). Trypsin and fetal bovine serum (FBS) were purchased from Hyclone (Logan, UT, USA). Ethylenediaminetetraacetic acid (EDTA), 2′,7′-dichlorofluorescin diacetate (DCFH-DA), and paraformaldehyde were purchased from Serva (Heidelberg, Germany). DMEM and RPMI 1640 culture medium were purchased from Gibco (Waltham, MA, USA). Furthermore, 3-(4,5-dimethyl-thiazol-2yl)-2,5-diphenyltetrazoliumbromide (MTT), giemsa, mowiol, ophthaldialdehyde, and poly(*2*-hydroxyethyl methacrylate) (poly-HEMA) were purchased from Sigma-Aldrich (St. Louis, MO, USA). Cleaved PARP1 p25 Rabbit mAb (A19612), β-Actin Rabbit mAb (AC026), and HRP Goat Anti-Rabbit IgG (AS014) were purchased from Abclonal (Woburn, MA, USA). Fluid thioglycollate medium and soybean-*c*asein digest medium were purchased from Sigma-Aldrich (St. Louis, MO, USA). Trichostatin A (TSA) war purchased from SelleckChem (Houston, TX, USA), a 50 μM solution of TSA in DMSO was used in the study. Deuterated water was purchased from Komponent-Reaktiv (Moscow, Russia). Methylene Blue was purchased from Sigma-Aldrich (Darmstadt, Germany). Human serum albumin (HSA) was purchased from Paneko (Moscow, Russia). Water was purified in a Millipore Milli-Q Plus system (Darmstadt, Germany).

### 2.2. Methods

#### 2.2.1. Photophysical Properties of XL

##### Fluorescence and Singlet Oxygen Measurements

Fluorescence and singlet oxygen luminescence registration experiments were performed with a FluoTime 300 spectrometer (PicoQuant GmbH, Berlin, Germany) equipped with NIR PMT Module H10330-45 (Hamamatsu, Japan). Fluorescence quantum yield of XL was calculated using tetraphenylporphyrin TPP as a standard (0.12) in acetonitrile [15]. The OD of sample and standard were matched (≈0.05 per cm) at the excitation λ. As a refence standard for NP–XL, singlet oxygen determination Methylene Blue in D_2_O was used Φ_Δ_ = 0.52 [16]. Both fluorescence and singlet oxygen quantum were calculated according [17]. Samples both of XL and NP–XL in D_2_O were prepared with matched optical density (<0.1 per cm) at the excitation λ. Xe-lamp was used as the excitation source (λ_ex_ = 675 nm). The light scattering of NP–XL samples was accounted for using NP without XL compound. Measurements were performed using a quartz cuvette with a 1 cm optical path.

##### Flash Photolysis

The transient absorption spectra of radical intermediates were measured using a conventional flash photolysis setup (LLC “MELZ”, Moscow, Russia) (optical path length 20 cm, excitation performed through red filters > 550 nm, 80 J/20 μs). Signals were recorded using a PMT-38 photomultiplier (LLC “MELZ”, Moscow, Russia) at 380–740 nm.

##### Laser Flash Photolysis

Laser pulse photolysis was performed on an LKS 80 (Applied Photophysics, Leatherhead, UK) laser pulse photolysis setup. The third harmonics of a Nd-YAG laser (Brilliant B, Quantel, LUMIBIRD, Cournon-d’Auvergne, France) was used for excitation. Excitation λ in the range 420–600 nm was tuned by an OPO (MagicPrism, OPOTEK Inc., Carlsbad, CA, USA). Kinetics of transient species generated by a 5 ns laser excitation pulse were registered by absorbance changes in the spectral range of 400–750 nm using a 150 Xe arc lamp with 50-fold beam overdrive and 1.5 ms capacitor discharge. The detection system was equipped with a 600 MHz oscilloscope (Agilent Infiniium 10 GS/s) an R928-type PMT (Hamamatsu Photonics, Shizuoka, Japan). The kinetic data were processed through global analysis by fitting the kinetic traces over the whole range of the registration λ with a multiexponential Equation (1):ΔA_λ_ = ∑ΔA_λi_exp(−t/τ_i_),(1)
where ∆A_λ_ is the overall difference absorbance at the registration λ, ∆A_λi_ is the absorbance of the i-th transient species at the registration λ, and τ_i_ = 1/k_i_ is the lifetime of the i-th transient species, with τi and ∆A_λi_ being the fitting parameters. The accuracy in the lifetime determination was 15%. The sample and the standard OD were matched at the excitation λ (419 nm). The triplet state quantum yield was evaluated with the accuracy ±20%.

##### XL–HSA Complex Formation

The XL emission spectra were recorded at different HSA concentrations (0.05, 0.125, 0.25, 0.5, 0.75, 1.0, 1.5, 2.0, 4.7, 6.8 (×10^−5^ M)). The binding constant was calculated according to Equation (2):F − F_0_ = F_max_ × [HSA]/(K_d_ + [HSA]),(2)
where F_0_, F and F_max_ are the XL fluorescence intensity in the absence of HSA, at different HSA concentrations and at the HSA concentration where complete complex formation takes place, correspondingly. K_d_ is the dissociation constant which is inverse proportional to the binding constant K_b_.

#### 2.2.2. XL Binding to PVA in Solution

XL binding to PVA was studied in a model homogeneous system consisting of 1% aqueous PVA solution containing 0.33 vol.% DMSO. For this purpose, 10 μL of a stock XL solution in DMSO (*C* = 2 mM) was added upon stirring to 3 mL of DMSO, water, or 1% aqueous PVA solution. UV-Vis absorption spectra of the samples were recorded using a HACH DR4000V spectrophotometer (Loveland, CO, USA) and fluorescence spectra were obtained from a Perkin Elmer LS-50 luminescent spectrometer (Perkin Elmer, Beaconsfield, UK). All the measurements were made at room temperature in 10 mm quartz cells at an excitation λ of 500 nm and a monochromator slit width of 6 nm.

#### 2.2.3. Synthesis and Scale-Up of NP–XL

XL-loaded PLGA NPs were formulated by a single emulsion technique, as described previously [6]. Briefly, a mixture of 5 mg of XL and 50 mg of PLGA was dissolved in chloroform and added dropwise to the 25 mL of 1% *w*/*v* PVA aqueous solution. The suspension was stirred for 20 min, followed by ultrasonic homogenization (Labsonic U. B. Braun, Melsungen, Germany) five times for 40 s in ice bath. The organic solvent was then rapidly removed on a vacuum rotary evaporator (IKA HB10, Staufen, Germany) at +30 °C. The obtained sample was separated by centrifugation for 30 min at 14,000 rpm and +4 °C (Beckman J221, Brea, CA, USA). The supernatant was decanted, and pure deionized water was poured into the centrifuge tube, followed by vigorously stirring to wash the NPs pellet. Finally, 10% (*w*/*v*) D-mannitol was added, and the mixture was freeze-dried by lyophilization (Alpha 2–4, Martin Christ GmbH, Harz, Germany).

Scale-up production of XL encapsulated nanoparticles was similar to the standard lab-scale process. To produce a threefold scale-up batch, three mixtures of XL (5 mg) and PLGA (50 mg) were dissolved and added dropwise to a corresponding 1% PVA aqueous solution. Each suspension was homogenized and the resulting emulsions were combined to form a uniform mixture. The organic solvent evaporation and mixture lyophilization were carried out as described above.

To produce sixfold scale-up batch, the initial amount of the substance, polymer, and excipients were increased twofold. Any further procedures were similar to that of the threefold scale-up batch preparation.

Blank NPs were synthesized according to threefold scale-up batch, except for XL addition.

Long-term stability of NP–XL was evaluated while storing the samples in closed glass vials without access to light at +4 °C for 6 month. Samples were withdrawn after month 1, 3, and 6, diluted with water, and analyzed for particle size, polydispersity index (PDI), ζ-potential, drug loading (DL), and encapsulation efficiency (EE).

#### 2.2.4. Characterization of NP–XL

##### Size, ζ-Potential-Potential, and Polydispersity Index Measurement

Particle size, ζ-potential-potential, and PDI were determined with ζ-potential sizer Nano ZS ZEN 3600 analyzer (Malvern Instruments, Malvern, UK) as was described previously [18]. The responses were measured in triplicates.

##### HPLC Analysis of XL

A reverse-phase liquid chromatography method was developed to quantify XL using a chromatographic system including 1525 binary pump (Waters, Milford, MA, USA) and 2487 UV-Vis detector (Waters, Milford, MA, USA). Chromatographic analysis was performed using Breeze 3.0 system (Waters, Milford, MA, USA) and Jupiter C18 analytical column with the dimensions of 150 mm × 4.6 mm ID × 3 µm (Phenominex, Washington, DC, USA). The flow rate was set at 1 mL/min with UV detection at 407 nm. The mobile phase consisted of acetonitrile: 0.1% trifluoroacetic acid solution (70:30 *v/v*). Linear gradient from 70% to 100% B was applied for 5 min, followed by isocratic elution at 100% B for 5 min. The mobile phase was filtered through a 0.45 µm nylon membrane filter (Phenomenex, Washington, DC, USA). The sample injection volume was 10 µL and the analysis was carried out at ambient temperature.

##### DL and EE Measurement

DL and EE were analyzed by HPLC as described above. To determine DL of XL, 5 mg of NPs were accurately weighed and suspended in 20 µL of DMSO and HPLC mobile phase to a final concentration of 1 mg/mL. Ten µL of the resulting solution were injected into the HPLC system. To determine EE of XL, 5 mg of NPs were accurately weighed and resuspended in distilled water (4 mL) and centrifuged for 30 min at +4 °C and 15,000 g. The supernatant was removed, and the procedure was repeated. Then, the precipitate was transferred into a 5 mL volumetric flask with mobile phase added to 5 mL. Ten µL of the resulting solution were injected into the HPLC system.

DL (Equation (3)) and EE (Equation (4)) were calculated as follows:DL, % = (weight of remained drug in the particles/weight of particles) × 100(3)
EE, % = 100 − ((total amount of drug − actual amount of drug in precipitate)/total amount of drug)) × 100(4)

##### Morphological Study

Shape and surface morphology of NP–XL were analyzed with scanning electron microscopy (SEM) and transmission electron microscopy (TEM). A suspension NP–XL in distilled water was dropped onto the surface of a formvar-coated copper grid (Gilder Grid Square 200 Mesh), followed by the evaporation of the solvent to prepare sample for TEM. Then, 10 µL of UranyLess solution (Electron Microscopy Sciences, Hatfield, PA, USA) was also dropped onto the surface of a formvar-coated copper grid and incubated for 20 s to enhance the contrast of the TEM images. The grid was then washed twice in deionized water. To prepare the sample for SEM, a drop of NP–XL water suspension (1 mg/mL) was placed on a double-sided carbon tape, respectively, and dried for 30 min. Samples were coated with a gold layer. TEM analysis was performed on a JEM-1400 microscope at 120 kV (JEOL, Tokyo, Japan). SEM analysis was performed on a JSM-6510LV microscope (JEOL, Tokyo, Japan).

##### Determination of PVA Residual Content

The residual PVA content was quantified using a colorimetric method that relies on the formation of a colored complex between the adjacent hydroxyl groups of PVA and iodine molecules [19]. Briefly, 2 mg of NPs were treated with 1 mL of 0.5 M NaOH for 15 min at +60 °C. Samples were then neutralized with 900 µL of 1 M HCl and their volumes were adjusted to 4 mL with distilled water. Three mL of a 0.65 M boric acid solution, 0.5 mL of 0.05/0.15 M I2/KI solution, and 1.5 mL of distilled water were added to neutralized samples and incubated for 15 min. Finally, the absorbance of the samples was measured at 690 nm with UV-Vis spectrophotometer Spectronic Helios Alpha (Thermo Fisher Scientific, Waltham, MA, USA). A standard plot of PVA was prepared under the same conditions.

##### Fourier Transform Infrared Spectroscopy (FTIR)

FTIR spectra were recorded for XL, PLGA, and NP–XL using FTIR spectrometer (FTIR EQUINOX 55, Brucker, Chicago, IL, USA). NPs were lyophilized prior to FTIR analysis. The samples were prepared using micronized KBr and scanned from 400 cm^−1^ to 4000 cm^−1^.

##### X-ray Diffraction (XRD)

An XRD study of XL, NP–XL, and excipients was carried out using an X-ray diffractometer of local design [20], supplied with the optical focusing of the X-ray beam and a linear position sensitive X-ray detector. A fine-focus sealed Cu X-ray tube with Ni β-filter was used as an X-ray source. The cross section of the X-ray beam in the plane of the sample was ~4 × 0.15 mm^2^. An X-ray detector was installed with an inclination toward the sample at ~20°, the sample to detector distance was ~92 mm, and the width of the detector window was limited to 4 mm by the slit overlay in order to diminish the smearing of XRD patterns. X-ray detector was calibrated with diffraction lines of corundum and poly(3-hydroxybutyrate). XRD intensity was measured in the transmission mode, corrected for background scattering, and plotted as a function of diffraction vector module S = 2sinθ/λ (2θ − scattering angle, λ−X-ray wavelength, equal for CuKα radiation to ~0.1542 nm).

##### Differential Scanning Calorimetry

The thermophysical characteristics of the samples were measured on a DSC 204 F1 Phoenix^®^ (Netzsch, Waldkraiburg, Germany) differential scanning calorimeter with a heating rate of 10 K/min in an inert atmosphere of Ar. The experiments included only one heating stage, since we sought to characterize the morphology of the initially obtained compositions, and not to “erase their memory” or thermodynamically balance their primal structure, which is essential in the study of complex biopolymer compositions.

#### 2.2.5. Modeling of XL Release Kinetics

The drug release data were obtained previously [6] and modeled using the Excel^®^ (Microsoft, Santa Rosa, CA, USA) Add-In DDSolver.

#### 2.2.6. Sterilization

##### Gamma Irradiation

Different batches of NPs (10 mg) were transferred to 10 mL glass vials. Samples were γ-irradiated using a ^60^Co source (GUT-200M; Research Institute of Technical Physics and Automation, Moscow, Russia) at doses of 10, 15, and 25 kGy. Each cumulative dose was calculated based on varied dose rate (Gy/s) and exposure time (h).

##### Membrane Filtration and Sterility Testing

Ten samples were filtered through a 0.22 mm nylon membrane filter (Millipore, MA, USA). Resultant filtrates were transferred to autoclaved culture media (thioglycolate and casein soya) in conical flask. Incubation was conducted for one week at +37 °C for thioglycolate media and at room temperature for casein soya media. Microbial growth was determined by visual examination (changed color or turbidity).

#### 2.2.7. Cell Culture

The cell lines, MDA-MB-231 (human breast adenocarcinoma) (ATCC, Manassas, VA, USA), BJ-5TA (human skin fibroblasts) (ATCC, USA), and HeLa trichostatin-induced (human cervical adenocarcinoma, HeLa TI) cells, were maintained in DMEM, K562 (human myelogenous leukemia) were maintained in RPMI 1640 supplemented with 10% fetal bovine serum and gentamycin (50 µg/mL). Cells were grown in plastic 25 cm^2^ cell culture flasks at +37 °C in humidified atmosphere containing 5% CO_2_ (Sanyo, Osaka, Japan). Cells were seeded before reaching 80% confluence by detachment with trypsin/EDTA solution.

The experiments including human and animal tissues or cell cultures were conducted according to the guidelines of the Declaration of Helsinki and approved by the Institutional Ethics Committee of N.M. Emanuel institute of biochemical physics (protocol code 29 approved 16 February 2023).

#### 2.2.8. HeLa TI Cell-Based Assay

HeLa TI cells were seeded in 24-well plates (20 × 10^3^ cells per well) 24 h before the experiment and incubated under standard conditions. At day of the experiment XL, NP–XL were added to cells in duplicate. Final concentrations of XL in both cases were 0.25, 0.5, and 1 μM. For negative control, the cells were treated with blank NPs in the same concentration as with XL, and DMSO (0.1%), as vehicle control. TSA (0.25 μM), a histone deacetylase inhibitor, was used as a positive control of the reactivation of the epigenetically repressed *GFP* gene in HeLa TI cells. Afterwards, treatment cells were irradiated with 660 nm light-emitting diodes for 20 min with a 25 mW/cm^2^ power LED and then incubated for 24 h. Then, medium was changed to fresh and, after 48 h, flow cytometry was performed using BD FACSCanto II cytometer (488 nm laser, 530/30 filter, FITC). The results were interpreted using FACSDiva software (Version 6.1.3).

#### 2.2.9. Cytotoxic Activity (2D)

MDA-MB-231 and BJ-5TA cells were seeded in 96-well plates (5000 cells per well) 24 h before the experiment and incubated under standard conditions. K562 cells were seeded into wells (5000 cells per well) just after the XL or NP–XLs samples were added into 96-well plates. XL or NP–XLs were added in triplets in the concentration range 0.016–1 µM (according to XL concentration). To determine phototoxicity, cells were irradiated with 660 nm light-emitting diode for 20 min with a 25 mW/cm^2^ power LED and then incubated for 72 h. Cell survival was determined using standard MTT assay [21]; 50 µL MTT in DMEM/RPMI 1640 (1 mg/mL) was added into each well. After cell incubation for +37 °C, the medium was removed and precipitated formazan crystals were dissolved in 100 µL DMSO. Following this, the absorption intensity of formazan was measured at 540 nm on a microplate reader. Cell viability was determined as the percentage of untreated controls. The responses were measured in triplicates.

#### 2.2.10. Growth of MDA-MB-231 Spheroids (3D) and Cytotoxic Activity

MDA-MB-231 cells were cultured in DMEM supplemented with 10% FBS and 50 μg/mL gentamycin. Cells were maintained at +37 °C, a humidified incubator containing 5% CO_2_ (Sanyo, Japan). Poly-HEMA was adsorbed to 96-well plate in the concentration of 12 mg/mL in 95% ethanol and incubated at room temperature overnight. After the surface coating, MDA-MB-231 cells were seeded at density 2 × 10^3^ cells in 96-well plate and allowed to form spheroids for 24 h. The next day, cells were treated with XL or NP–XL in the concentration range 0.016–1 μM (according to XL concentration). After 2 h of treatment, cells were irradiated with 660 nm light-emitting diode during 20 min with the 25 mW/cm^2^ power LED and then incubated during 24 h. Cell survival was determined as described above.

#### 2.2.11. ROS and XL and NP–XL Subcellular Localization

The intracellular XL and NP–XL localization in MDA-MB-231 cells was analyzed by confocal laser scanning microscopy (CLSM). A total of 2 × 10^4^ (1 mL/well) cells were seeded on the coverslips in 24-well plates for 24 h. The next day, cells were incubated for 2 h in serum free media and treated with 100 nM XL or NP–XL for another 2 h, rinsed three times with PBS, and stained with 2.2 µM hoechst 33342. After treatment, cells were stained with 20 µM DCFH-DA and 10 µM MitoSox Red for 20 min in the dark and irradiated with a 660 nm LED with 25 mW/cm^2^ power for 20 min. The slides were rinsed, embedded in mowiol, and analyzed immediately by CLSM.

#### 2.2.12. Western Blotting

500,000 MDA-MB-231 cells were seeded into 60 mm Petri dish and allowed to adhere overnight. Next, cells were treated with 100 nM XL or NP–XL during 2 h, rinsed three times with media, and irradiated with a 660 nm LED with 25 mW/cm^2^ power for 20 min. After 3 h of incubation, cells were harvested, washed twice with ice-cold PBS, and incubated in 100 μM lysis buffer (25 mM Tris-HCl, pH 7.4, 1% sodium deoxycholate (*w*/*v*), 1% Triton X-100 (*v*/*v*), 5 mM EDTA, 150 mM NaCl, 1 mM phenyl methyl sulphonyl fluoride, 0.1% sodium dodecyl sulfate (SDS) (*w*/*v*)) for 30 min on ice. The cell lysates were centrifuged at 12,000× *g* for 20 min at 4 °C. The supernatant was collected, and the protein concentrations of the supernatant was measured using a Bicinchoninic acid protein assay kit. The extracts were applied to 12% and 15% SDS polyacrylamide gel electrophoresis, the proteins were transferred onto a polyvinylidene difluoride membrane, and blocked after with a 5% enhanced chemiluminescence (ECL) blocking agent in TBST buffer (12 mM Tris–HCl, 75 mM NaCl, and 0.05% Tween 20) for overnight at 4 °C. After triple washing in TBST buffer, the primary cleaved PARP rabbit monoclonal antibody and β-actin rabbit monoclonal antibody as control were applied on membrane and incubated for 1 h at room temperature. Cell lysates were washed thrice in TBST buffer and incubated with horseradish peroxidase-conjugated anti-rabbit secondary antibodies for 1 h. After washing samples three times with TBST buffer, we revealed the antigen–antibody interactions with composition of 1:1 luminol Peroxide Buffer. The chemiluminescence signal was analyzed with ChemiDoc Imaging System (BioRad, Hercules, CA, USA).

#### 2.2.13. Data Analysis and Statistical Evaluation

Statistical analysis was performed using an independent sample *t*-test with OriginPro 7.5 software. The normality of FACS data distribution was assessed with the Kolmogorov–Smirnov test. The significance of difference in the effects of compounds relative to vehicle control in HeLa TI cell-based assay was analyzed using one-way ANOVA with Dunnett’s multiple comparisons test. The difference between the groups was considered significant at *p* ≤ 0.05.

## 3. Results and Discussion

### 3.1. Photophysical Properties of XL

XL absorption spectrum showed characteristic pyropheophorbide α pattern with intense band in the region of 380–420 nm (Soret band) and several bands in the region of 500–700 nm (Q-band; Figure 1b) [22]. Regarding fluorescence spectrum, XL exhibited maxima at 672 nm with a shoulder-peak at 720 nm. The fluorescence decay kinetics was monoexponential with the lifetime of 6.9 ns (Figure 1c). Fluorescence quantum yield of XL in acetonitrile was 0.36.

Next, we performed flash photolysis experiments to evaluate the XL intersystem crossing process. After XL (3.0 × 10^−7^ M) photoexcitation (λ_ex_ > 550 nm) in degassed acetonitrile solution the new absorption bands corresponding to the triplet state absorption emerged in the 440–580 nm region with maximum at 460 nm. At the same time, we observed intense photobleaching of the ground singlet state at Soret band which coincided with the singlet absorption spectrum (380–430 nm) and the most intensive Q-band (630–680 nm) (Figure 1d). The triplet kinetics decay had the rate constant of *k*_T_ = 2.5 × 10^3^ s^−1^ (Figure 1d, Inset).

The experiment of the triplet state quenching by oxygen resulted in the rate constant 2.6 × 10^9^ M^−1^·s^−1^. Obtained value was close to diffusional rate constant in acetonitrile [23] including 1/9 spin-statistical factor (2.0 × 10^9^ M^−1^·s^−1^) indicating that quenching process is diffusion controlled.

The efficiency of intersystem crossing processes in XL was evaluated with triplet state quantum yield determination using laser flash photolysis setup. As a reference standard, we used ZnTPP solution in toluene (Φ_T_ = 0.83, ε_T_ = 7.4 × 10^4^ M^−1^·cm^−1^ at 470 nm [24]. We observed the linear dependence between laser excitation energy and the initial triplet state absorption for both standard and sample (Appendix A).

The triplet state extinction for XL was evaluated using singlet depletion method according to Equation (5):ε_T_ = ε_S_ (ΔOD_T/_ΔOD_S_),(5)
where ΔOD_T_ and ΔOD_S_ are differential optical densities after the laser pulse for XL triplet state at 460 nm and XL singlet state at 410 nm, respectively; ε_S_ is molar extinction coefficient of the singlet state at 410 nm (2.05 × 10^5^ M^−1^·cm^−1^). The resulting value of ε_T_ was 5.1 × 10^4^ M^−1^·cm^−1^ (at 460 nm). According to the slope values from Appendix A, XL triplet state extinction coefficient and the method described [24], the estimated XL triplet quantum yield was 0.52. The flash photolysis experiments results provided XL triplet state information and the intersystem crossing process efficiency which conformed with the singlet oxygen quantum yield value in DMSO (Φ_Δ_ = 0.62) evaluated earlier [25] that indicated the efficient energy transfer process.

Overall, these results confirm previously reported data describing high ROS-based anticancer activity of the compound [6].

### 3.2. XL–HSA Complex Formation

Earlier, we observed the high photocytotoxicity of XL determined by ROS formation [6]. However, in aqueous solutions XL forms aggregates [26] and lack fluorescence and ability to generate singlet oxygen (see Section 3.12). In order to reveal the base of XL photocytoxicity, we studied the changes in photophysical properties of XL molecule in complex with HSA. Binding with HSA is important to evaluate the photosensitizer delivery into cells.

Upon the addition of albumin to the XL buffered solution, the fluorescence increased drastically (Figure 2a) due to incorporation of the compound molecule into protein structure accompanied by the enhancement of XL rigidity. As a result, the internal conversion process diminished, and fluorescence amplified. The resulting binding constant value (K_b_ = 2.0 × 10^5^ M^−1^) indicated rather strong binding ability of XL molecule.

We studied the triplet state of XL-HSA complex using flash photolysis technique. The experiment was performed in the presence of albumin excess in degassed phosphate-buffered solution. The differential absorption spectrum (Figure 2b) included the bleaching of XL singlet state in Soret band (400–420 nm) and the most intensive Q-band (640–680 nm) along with the triplet absorption bands at 460–580 nm region. Along with the HSA concentration increase, the XL triplet absorption became more intense (Figure 2b, Inset A), increasing the dye monomolecular form concentration bound with the protein. Upon complex formation, the internal conversion process of the dye diminished, and both fluorescence and intersystem crossing increased.

The rate constant of triplet state decay was 2.6 × 10^3^ s^−1^ and it was unchanged in the presence of HSA (±10%). The bimolecular rate constant of triplet state oxygen quenching was 6.0 × 10^8^ M^−1^s^−1^ (Figure 2b, Inset B). The obtained value was three times less compared to the typical diffusional rate constant in aqueous solution (≈1.9 × 10^9^ M^−1^s^−1^) [25]. This indicated the steric hindrance for the oxygen molecule contacting XL bounded to protein. The analysis of XL–HSA photophysical properties showed that the presence of the protein leaded to both XL fluorescence and triplet state formation increment due to internal conversion decrement. The absence of singlet oxygen generation in XL aqueous solutions and observed dye photocytotoxicity in the cells could be explained with the XL triplet state formation in complex with HSA.

### 3.3. XL Binding with PVA

The hydrophobic behavior of chlorins and chlorophyll-based PS cause self-aggregation in aqueous solution reducing their fluorescence quantum yield, triplet state, and singlet oxygen generation, consequently diminishing their photosensitizing activity [27]. To address this limitation, among different approaches, non-covalent binding of PS with PVA is applied. PVA is a biocompatible, non-degradable, non-toxic polymer, widely applied as excipient, including PLGA nanoparticles [26,27]. Based on this, we decided to evaluate the potential influence of XL binding to PVA on photoactivity and to analyze the optimal XL: PVA ratio.

In order to reveal the possibility of XL binding to PVA in solutions, absorption and emission spectra of XL in DMSO, water, and 1% aqueous solution of PVA were compared. From the graphs shown in Figure 2c, we concluded that XL in water, compared to DMSO, was characterized by a twofold extinction decrease and significant broadening of the absorption bands, as well as a strong fluorescence quenching resulting from self-association of the hydrophobic XL molecules in the polar media [26]. In the presence of PVA, we observed a partial recovery of the shape and intensity in both absorption and emission bands of XL, indicating XL solubilization in the monomolecular fluorescent form upon binding with the polymer. The possible mechanism of XL solubilization could be associated with the formation of micelle-like structures in aqueous PVA solution due to the amphiphilic nature of the partially hydrolyzed polymer [28].

The XL fluorescence increased along with the PVA concentration, and we also observed a gradual bathochromic shift of emission maximum from 665 to 671 nm (Figure 2d) indicating an equilibrium shift in the XL monomer/aggregate system towards the monomeric form. A quantitative assessment of the XL binding to PVA indicated the maximum solubilization efficiency at the component ratio of 1 mg XL per 3 g PVA (Figure 2d, insert).

Thus, binding of XL molecules to PVA in aqueous medium resulted in increasing of the photophysical activity of the photosensitizer due to solubilization at the optimal component ratio of 1 mg of XL per 3 g of PVA. Thus, our results provide a rationale for PVA application as an emulsion stabilizer for further synthesis of nanoparticles.

### 3.4. Synthesis and Scale-Up of NP–XL

We used a single emulsion technique to formulate NP–XL because it is suitable for the encapsulation of hydrophobic substances [6,18]. Emulsification is a crucial step in PLGA nanoparticles synthesis via single emulsion technique. We used ultrasound homogenization (UH) to yield emulsion of NPs (Figure 3). Ultrasound impact induce acoustic cavitation, disrupting oil droplets and facilitating the formation of stable O/W emulsions with small-size droplets [29]. The efficacy of acoustic cavitation *is* controlled by sonication power. The volume of sample suspension also affects the emulsion parameters: the larger the volume, the less the efficiency. When scaling, *we* increased the number of PLGA–XL mixtures, while the volume of each mixture remained the same.

We performed the threefold scaling by emulsifying three PLGA–XL mixtures, added dropwise to three PVA solutions (Figure 3). After UH, all the solutions were mixed to evaporate the organic solvent. To perform sixfold scaling, we doubled the initial amounts of PLGA, XL, and PVA solution, followed by sonication and evaporation as described above.

Table 1 shows the main parameters of NP–XL after synthesis and scale-up process. The non-scaled batch (1-fold) had high levels of XL loading, showing high values of DL and EE. The size and PDI values were appropriate for drug delivery via EPR-effect as the pore size in the capillaries of solid tumors can be up to 300 nm [30]. The value of ζ-potential indicated the physical stability of nanosuspensions due to sufficient electrostatic repulsion between individual particles.

The threefold scaling-up resulted in a slight increase of all the parameters except ζ-potential and EE compared to non-scaled batch. The decrease in the ζ-potential value probably indicated shielding of surface carboxyl groups of PLGA with the substance or PVA [31]. The EE value was also slightly decreased compared to non-scaled batch, which can occur due to the more pronounced interaction of XL with PVA and the migration of the former to the particle surface [32,33]. Also, longer evaporation time led to XL release and slower NPs formation, decreasing EE value.

Based on increased values of size and PDI, we concluded that the sixfold scaling resulted in particles agglomeration. Insufficient sonication power had to be adjusted in order to provide appropriate treatment of increased volumes and loadings. Thus, we utilized the threefold scaling-up process to formulate sufficient amounts of NP–XL for further experiments.

Next, we confirmed the stability of XL after encapsulation by HPLC (Figure 4e). The chromatograms of XL and NP–XL had similar patterns—lack of new peaks of degradation products or major and minor peaks shift. Our results indicate that XL remained stable after encapsulation process.

After six months of NP–XL storage in a closed glass vials without light access at +4 °C, freeze dried NPs were stable, and were characterized with insignificant changes in size, PDI, ζ-potential, DL, EE (*p* > 0.05; Appendix A), and retained visual appearance (intact green pellet). We observed no changes in visual description (intact green pellet) indicating good stability profile of NP–XL up to six months.

### 3.5. Morphological Analysis

We performed the morphological examination of NP–XL by SEM and TEM analysis (Figure 4a,b). We observed spherical shape and smooth surface of the NP–XL with relatively low size dispersion.

### 3.6. Determination of PVA Residual Content

Previously Sahoo and coauthors showed that a fraction of PVA associated with the surface of the PLGA nanoparticles significantly affected the physical parameters (surface hydrophobicity, size, and DL) and the cellular uptake of NPs [34]. These findings emphasized the importance of PVA residual content analysis. We evaluated this value by a colorimetric method that relies on the formation of a complex between the adjacent hydroxyl groups of PVA and iodine molecules. Next, we determined the concentration of the complex applying spectrophotometry at 690 nm [19].

We compared the residual content of PVA after nanoparticle synthesis in blank NPs and NP–XL (Table 2). We also analyzed the influence of centrifugation on PVA concentration.

The results showed that centrifugation significantly (*p* < 0.05) reduced the PVA concentration on the surface of PLGA nanoparticles. PVA concentration in NP–XL was slightly increased; however, no significant difference was revealed. Importantly, complete PVA removal can lead to aggregation of the NPs resulting rapid precipitation and impair intravenous administration. Therefore, excessive removal of PVA (more than 70%) is unreasonable.

Thus, we concluded that centrifugation was an efficient technique to reduce the amount of residual PVA. The presence of an encapsulated drug lacked significant influence on residual PVA concentration.

### 3.7. Fourier Transform Infrared Spectroscopy

Chemical interactions between the drug and the polymer were examined by FTIR spectroscopy (Figure 4c).

PLGA showed characteristic peaks at 2996–2949 cm^−1^ (C-H stretches), 1747 cm^−1^ (C=O stretch), 1383 and 1084 cm^−1^ (asymmetric and symmetric C-C(=O)-O stretches, respectively), and 868–748 cm^−1^ (C-H bends) [35]. XL spectrum revealed complicated FTIR pattern characterized to chlorines with several regions [36]. Furthermore, the 680–900 cm^−1^ region included both in-plane and out-of-plane vibrations. The strongest peak at 671 cm^−1^ belonged to out-of-plane vibration that was primarily due to methine bridge bending of the pyrrole rings [37]. In the 900–1650 cm^−1^ region, mainly in-plane vibrations appeared. We also observed characteristic peaks at 1686 cm^−1^ (C=O stretch), 2958–2867 cm^−1^ (C-H stretching), and broad peak at 3389 cm^−1^ (O-H stretching). The NP–XL spectrum was similar to that of PLGA, except the fused peak at 2946 cm^−1^ (C-H stretching) and the slightly shifted broad peak at 3271 cm^−1^ (O-H stretching). The latter could be attributed to XL groups, confirming encapsulation. The peak at 3271 cm^−1^ (O-H stretching) also could be attributed to water molecules presented on the NP–XL surface. Previous studies revealed that lyophilized nanoparticles still contained 3.5% of residual moisture [38], which is acceptable and results in only non-significant PLGA degradation [39]. The absence of certain characterization peaks of XL in the NP–XL spectrum is due to the stronger signals of PLGA chemical bonds. These signals mask the FTIR signals of the encapsulated XL, as the latter is present in smaller quantities on the surface of the nanoparticles compared to the polymer.

Thus, we observed a lack of new peaks in the NP–XL spectrum indicating XL encapsulation into PLGA nanoparticles via physical interactions. No new chemical bond formation was observed.

### 3.8. DSC Studies

DSC is a useful tool for polymers composites analysis aimed to investigate [40]:interactions between polymer and drugs,crystalline morphology of the polymer and the state of encapsulated drug (solid or molecular dispersion),intermolecular interactions between drugs and pharmaceutical adjuvants.

Figure 4d shows the DSC curve of PVA exhibited a glass transition temperature (Tg) value between 48.3 and 55.0 °C related to the amorphous part of the material. The second endo peak of PVA referred to the melting of the crystalline phase occurring at 192.9 °C (ΔH_m_ −46.1 J/g), which was in line with typical results of PVA with hydrolysis degree of 87–90% [41]. The lower temperature of 164.4 °C may be attributed to the Tm of imperfect PVA crystallites. Equation (6) describes the melting point calculation [42]:T_m_ = T^0^_m_ × (1 − (2σ_в_)/(*l* × Δh))(6)

The depression of the melting point was related to the lamellar thickness *l*, fold surface free enthalpy σ_в,_ and the heat of melting per unit volume ∆h. The depression of the melting point of PVA could be explained by the formation of intermolecular hydrogen bonds of PVA, which hampered the normal folding of PVA chains, enhanced the fold surface free enthalpy, and reduced the lamellar thickness of PVA, hence, results in the depression of the melting point of PVA crystallites.

The DSC curve of D-mannitol exhibited sharp endothermic peak at 170.0 °C (ΔH_m_ −267.5 J/g), which was in line with previous results [43]. The DSC curve of PLGA showed the glass transition temperature of 44.7 °C followed by an endothermic event of relaxation at 45.0 °C [44]. Park et al. reported similar Tg values for PLGA 50:50 over the 45–50 °C temperature range [45]. The “relaxation” effects in PLGA and PLGA compositions illustrated by endothermic heat capacity peaks can be seen at temperatures above the Tg (Figure 4d (1, 4)). These peaks characterized the phenomenon of enthalpy relaxation which was associated with the mobility and the recovery of PLGA chains to their thermodynamic equilibrium status at a temperature above Tg.

The DSC curve of NP–XL composition exhibited two endothermic peaks at 147.9 °C and 160.0 °C, likely corresponding to the melting of XL. Compared to the endo-peaks of XL, they were shifted about 10° towards higher temperatures: from 128.9 °C and 141.5 °C with the total value of the melting at −48.1 J/g to 147.9 °C and 160.0 °C (ΔH*_m_* = −35.0 J/g), respectively, due to the influence of excipients: PVA and D-mannitol. This may indicate the physical interactions between them and XL. PLGA Tg value also shifted from 45.0 °C to 48.6 °C, which is in agreement with general properties of PLGA nanoparticles and occurs likely due to the encapsulation process [46]. In addition, the shift in PLGA Tg suggested weak interactions between the PLGA matrix and XL [47].

### 3.9. X-ray Diffraction Study

The XRD patterns of XL, PLGA, PVA, and D-mannitol and NP–XL are shown in Figure 5.

Pure PLGA exhibited diffuse X-ray scattering with broad intensive peaks at S ≈ 2.2 nm^−1^, corresponding to PLGA amorphous structures with a short-range order of atoms. The XRD pattern of XL, besides a diffuse peak at S ≈ 2.2 nm^−1^, showed a number of narrower peaks (the main peak lies at S ≈ 0.27 nm^−1^) indicating a more complex and ordered structure compared to the amorphous one. XRD pattern of PVA revealed low-ordered amorphous-crystalline structure. The XRD pattern of D-mannitol showed sharp diffraction peaks indicating the crystalline structure. The literature describes several crystalline forms of D-mannitol. While comparing the positions of diffraction peaks of D-mannitol in Figure 5 with the literature data, we concluded that the studied D-mannitol has the β-crystalline form [48].

XRD pattern of NP–XL showed intensive diffuse X-ray scattering with a pronounced peak at S ≈ 2.2 nm^−1^ indicating NP–XL amorphous structure. Besides diffuse scattering, a set of narrow low-intensity peaks present on XRD patterns XL–NPs indicating minor crystalline inclusions in these samples.

The profile of NP–XL diffuse X-ray peaks at S ≈ 2.2 nm^−1^ differs from that one for PLGA. We approximated diffuse X-ray scattering of NP–XL in the S range of 1–5 nm^−1^ by a sum of PVA and PLGA diffraction patterns taken in proportion of ~1:4. Based on the approximation, we confirmed that, besides PLGA, NP–XL also contain PVA.

Diffraction peaks of crystalline inclusions in NP–XL according to their positions could be attributed to D-mannitol in δ-crystalline form [48]. We determined the crystalline inclusion content in the NP–XL, which was 8%, via the ratio of the overall integrated intensity of crystalline diffraction peaks to the integrated intensity of the diffuse X-ray scattering; the average size of crystalline inclusions in NP–XL, determined by the analysis of diffraction line width, was 25–35 nm.

The XRD pattern of NP–XL lacked any diffraction peaks registered in XL XRD patterns, indicating that encapsulated drugs did not form inclusions with an ordered structure as in the XL substance. Based on the results, we concluded that the molecular-dispersed form or the form of small clusters was encapsulated in PLGA matrix XL.

### 3.10. Modeling of XL Release Kinetics

XL had biphasic release profiles from NPs in 0.01 M PBS media. We used 0.1% (*w*/*v*) Tween 80 to remove the sink limitation associated with XL high hydrophilicity [49].

We observed burst release during initial phase for 4 h—55% of XL (Figure 6a). After 4 h, a slow-release phase occurred—65% for 72 h. This pattern is typical for the release of hydrophobic drugs from PLGA nanoparticles [50]. Burst release is likely occurring due to the partial localization of the drug on the NP surface [51]. Kim et al. also proposed that water molecules penetrated into PLGA matrix affect the initial burst phase, reducing the drug–polymer interaction [50]. We observed a faster XL release at pH 5.0, relative to pH 7.4, characterized with release profile.

Next, we simulated the drug release kinetics to determine the limiting phase of the process, which corresponds to the main mechanism of drug release from the polymer matrix. We selected six conventional models to describe the XL release kinetics (Table 3). Here, the coefficient of determination R^2^ was a criterion for fitting the experimental data to the release curve.

Based on model fitting results, the Korsmeyer–Peppas model was best-fitted for describing XL release kinetics. The value of the exponent n in the Korsmeyer–Peppas equation determines the model of drug release: for spherical particles, the value n = 0.45 indicates release in accordance with Fick’s law; 0.45 < n <0.85 corresponds to anomalous transport; and n > 0.85 to the case of diffusion II (non-Fickian diffusion) [51]. In the case of spherical particles with n < 0.45, quasidiffusion proceeded by the Fickian mechanism occurred [52]. The latter case describes the n calculation results of n value obtained for NP–XL (Table 4). This type of transport indicates that the release of XL is mainly limited by diffusion, while the process of the swelling of the polymer matrix has little effect on the kinetics of release [53]. Our results corresponded previously reported data describing hydrophobic drug release mechanisms [18].

Thus, the approximation of XL release from NPs for the Korsmeyer–Pappas model had high values of determination coefficient. The value of n exponent indicated a Fickian diffusion mechanism and corresponded with the models of controlled release kinetics.

### 3.11. Sterilization

Steam sterilization (autoclave), electron beam technology, gamma irradiation, and sterile filtration are available for GMP production of parenteral PLGA formulations [54]. Gamma irradiation has significant advantages over the mentioned procedures: high penetration power, lack of sterilizing residues (e.g., water), better assurance of product sterility, and lower validation demands [55]. However, previous studies have reported that gamma irradiation caused degradation of PLGA [56]. We evaluated the stability of NP–XL after gamma irradiation at doses, indicated as efficient for sterilizing pharmaceutical products (Table 4) [57].

Particle size and PDI measured by DLS demonstrated that the applied doses of gamma irradiation (up to 25 kGy) significantly alter described parameters (Table 4). We also observed significantly reduced DL values in all samples. Regarding ζ-potential and EE, applied doses induced a slight decrease in the values of listed parameters. Thus, DLS studies revealed a high tendency to NPs’ agglomeration after irradiation at applied doses. The damage of PLGA molecular structure likely induces cross-linking of the polymer chains [58].

Also irradiation may cause formation of reactive radicals, which impair the stability of the encapsulated drug, reducing the DL values [59]. Thus, we evaluated the influence of gamma irradiation on the stability of XL loaded into PLGA NPs by HPLC (Figure 6b).

We observed a lack of changes in peak shape and retention time for NP–XL after irradiation at doses of 10 kGy, 15 kGy, and 25 kGy (Figure 6b). However, gamma irradiation resulted in significant increases of XL-summarized impurities of up to 14% (for 10 kGy and 15 kGy) and 15% (for 25 kGy). The peak of degradation product is well distinguished in chromatogram at 12.8 min. Our results indicated instability of NP–XL after irradiation at all the applied doses.

Based on these results, we concluded that gamma irradiation is unsuitable for NP–XL sterilization. And we used sterile filtration for further in vitro experiments.

Membrane filtration for PLGA nanoparticles sterilization can cause excessive NPs precipitation on membranes in case of sizes greater than 0.22 µm [46]. We evaluated the influence of membrane filtration on the NP–XL to ensure the suitability of this technique for sterilization. We analyzed the filtrated fraction of NP–XL by comparing XL concentrations in the suspension before and after the filtration procedure by HPLC. The filtrated fraction of NP–XL was determined to be 77%.

Sterility tests of NP–XL samples revealed a lack of biological contamination. Thus, our results confirmed that membrane filtration was an efficient method for NP–XL sterilization.

### 3.12. Singlet Oxygen Generation

It is important to determine whether the XL encapsulated into PLGA polymer is still capable to produce ROS., e.g., singlet oxygen is generated due to the energy transfer from the triplet state of photosensitizer to molecular oxygen (Equation (7)):Ps(T_1_) + ^3^O_2_ => ^1^O_2_ + Ps(S_0_)(7)

The phosphorescence of molecular oxygen could be registered at the NIR region of electromagnetic spectrum (λ_max_ = 1275 nm) (Equation (8))
^1^O_2_ => ^3^O_2_ + hυ(8)

Earlier, we showed that compound XL forms non-fluorescent aggregates in water [30]. XL in its aggregated state also does not form triplet states and no singlet oxygen luminescence was registered in D_2_O solution.

NP–XL demonstrated singlet oxygen generation capability both in absence of solvent (Figure 7b) and in D_2_O (Figure 7a). Quantum efficiency of the process was 0.47 in D_2_O. These results demonstrate that the encapsulation of XL into PLGA nanoparticles retained the singlet oxygen generation capability comparable with the value of Φ_Δ_ = 0.62 for XL in DMSO [26].

### 3.13. NP–XL Epigenetic Activity

One of the hallmarks of cancer cells is the aberrant level of epigenetic modifications, such as DNA methylation, histone acetylation, and methylation. Epigenetic abnormalities cause changes in the expression of oncogenes and tumor suppressor genes and, as a result, the tumor acquires a more aggressive phenotype. To normalize the balance of epigenetic modifications, chemical inhibition of the activity of epigenetic enzymes is used. Epigenetic modulators from the class of HDACs and DNMTs are currently approved for anticancer therapy. The use of epigenetically active agents is the perspective approach to improve the efficacy of photodynamic anticancer therapy. Epigenetic remodeling combined with PDT can represent effective mechanisms, leading to the development of anticancer immunity and potentiated anticancer effects [60]. The exposure of cancer cells to DNMT inhibitor 5-aza-2′-deoxycitidine and HDAC inhibitors vorinostat, Trichostatin A (TSA), and sodium butyrate leads to the reactivation of epigenetically repressed tumor suppressor genes, and increases the sensitivity of cells to PDT [61,62,63]. Compounds with both photodynamic properties and epigenetic activity are of great interest for antitumor therapy. Previously, no such drugs have been reported.

We analyzed the ability of XL and NP–XL to reactivate epigenetically repressed genes with HeLa TI cell-based test system, representing HeLa cells harboring the epigenetically silent GFP reporter gene. Previously, we verified the possibility of using this cell population as a model for the detection of epigenetically active xenobiotics [64,65]. Under an epigenetically active compound’s influence, *GFP* gene expression is reactivated, and GFP-positive (GFP+) cells can be easily detected by flow cytometry. The silencing of the *GFP* gene is supported by a number of epigenetic enzymes, such as HDAC1, DNMT3A, KMT1E, SUV420H1, SUV420H2, SUV39H1, and SUV39H2 и EZH2, which play an important role in the regulation of DNA methylation and histone modifications [66,67]. Thus, the drug-induced reactivation of the *GFP* gene indicates the ability of the compound to reactivate epigenetically repressed genes. We showed that, under the influence of XL, both free and encapsulated, there was a lack of increment in the proportion of GFP+ cells relative to vehicle (Figure 8b and Appendix A). The empty nanoparticles also did not cause the reactivation of *GFP* gene expression. TSA, positive control, caused a more than ninefold increase in the proportion of GFP+ cells.

Thus, XL lacked the ability to reactivate epigenetically repressed genes. This was the first study of the photodynamic therapy agents effects on the epigenetic system of transcription regulation.

Next, we analyzed the subcellular localization of XL and NP–XL in live MDA-MB-231 cells 2 h after NIR irradiation. The NP–XL predominantly displayed perinuclear localization (Figure 8a)—the distribution pattern indicated the endo-lysosomal accumulation, while more lipophilic XL demonstrated sparse homogenous distribution. Overlapping of mitoSox Red, DCF, and XL fluorescence, especially in the case of NP–XL, evidenced partial substance localization in or close to mitochondria, which agrees with earlier reports describing bacteriochlorins and phtalocyanins mitochondrial accumulation [68,69]. Weak intranuclear MitoSox Red fluorescence is explained by a time-dependent redistribution of the dye [70,71]. Both samples stimulated a prominent MitoSox Red fluorescence, but in the case of DCFH-DA, we observed relatively low fluorescence enhancement compared to control cells.

WB analysis revealed PARP cleavage stimulation, evidencing apoptosis stimulation, after irradiation in both cases, after treatment with XL and NP–XL.

The most probable mechanism of action of NP–XL (as well as XL) observed in vitro is the gradual release of XL inside cells. In the free state, the photosensitizer molecule acts mainly as a type 2 photosensitizer [72]–under the influence of light irradiation XL transforms into a triplet state, which was confirmed by flash photolysis experiments. Next, energy is transferred from the triplet state to the oxygen molecule, leading to the formation of an active form of oxygen (singlet oxygen).

These results demonstrated a high XL-8-NPs ability to internalize, redistribute along organelles, and, despite sustained release, stimulate oxidative stress and apoptosis at the same level as free XL-8 during the initial PDT phase.

### 3.14. In Vitro Cytotoxicity of XL and NP–XL

Previously, we showed that XL and NP–XL exhibited pronounced activity against HeLa cells [6]. The LED setup is really important and, generally, the light emission regime is selected in each individual case depending on many factors, where PDT effect and the lack of the influence on untreated cells are principal. Earlier, we found that 20 min light exposure is appropriate for XL and NP–XL PDT activity performing, which, at the same time, do not influence control cell survival in absence of the photosensitizers [6]. Aiming to expand the range of NP–XL applications, we evaluated cytotoxicity against triple negative human breast adenocarcinoma MDA-MB-231 and human myelogenous leukemia cell line K562. We selected MDA-MB-231 cell line (3D and 2D) as it has a high tumorigenic potential and is an often-used model of an aggressive triple negative breast cancer. K562 cell line was selected based on the prediction results of PASS-online service (Appendix A), where XL has the probability of being active against leukemia. We also evaluated XL and NP–XL activity against human foreskin fibroblast cells BJ-5ta representing control.

Figure 9 and Figure 10 show that XL and NP–XL performed up to 100% of cell viability in the absence of light, indicating absence of dark toxicity against 2D and 3D cultures, as well as against control cells. The irradiation dose was the same for 2D and 3D cultures. The light-irradiation mode was selected previously by the way it did not influence cell morphology and survival. The irradiation of wells containing XL and NP–XL stimulated a strong cytotoxic effect against K562 and MDA-MB-231 2D cultures (Figure 9b,c), with comparable IC50 values for both samples. The control cells also revealed high sensitivity to XL and NP–XL after irradiation (Figure 9d). We also noticed light toxicity elevation of XL high doses against control BJ-5ta cells in the dark, but the survival was too high to find IC50 value within the analyzed derivative concentration range (Figure 9d). At the same time, the NP–XL sample was safe under the same conditions (Figure 9d).

Next, we evaluated the toxicity of XL and NP–XL against MDA-MB-231 3D culture (Figure 9a and Figure 10). It is worth mentioning that in contrast with the 2D model (Figure 9b), MDA-MB-231 spheroids (Figure 9a) revealed less sensitive phenotype to XL and NP–XL after light exposure. Regarding the selective screening of drug cytotoxicity, 3D spheroids are becoming more and more widespread compared to 2D models. Three-dimensional spheroids reflect all kinds of intercellular interactions, as well as the interaction of cells with the extracellular matrix providing more adequate modeling of specific tissue architecture [73].

In present study, MDA-MB-231 spheroids had an average diameter of 230 μm and were reproducible with a coefficient of variation in spheroid diameter ≤ 8% (n = 8) on day one after seeding.

Figure 10 shows the representative images of irradiated, or non-irradiated MDA-MB-231 spheroids taken 24 h after the wells were supplemented with XL and NP–XL. The effect of XL and NP–XL was weaker compared to 2D model. This effect probably occurred due to tight cell packing with high density in spheroids, resulting in slower drug accumulation [74]. However, shorter incubation periods after light irradiation also should be taken into account—72 h in the case of 2D models and 24 h in the case of 3D models, which is explained by the difficulties of 3D culture support in plates. Thus, the IC50 values of MDA-MB-231 2D and 3D models barely could be compared directly, which did not diminish the significance high XL and NP–XL light-induced activity. Similar tendencies were described in the literature [75].

The images’ visual analysis let us assume more pronounced XL and NP–XL efficacy where we can see clear spheroid destruction (left and middle bottom images) (Figure 10). But, accordingly, the MTT analysis results confirmed these data-NP–XL were about two-times less toxic than XL, which was most likely explained by the gradual XL release from nanoparticles (Figure 9a).

Our results confirmed XL cytotoxic activity retaining after encapsulation into PLGA nanoparticles against both 2D and 3D tumor cell models. The nonspecific nature of ROS destructive effects also significantly influenced control BJ-5ta cells. Despite the slow gradual release of active substances, we observed similar toxicity levels against all cell lines, which may evidence at least XL active state support by PLGA matrix.

To summarize, our current and previous research results regarding new XL derivative and NP–XL formulation suggest that we should emphasize the promising PDT nature. As an effective photosensitizer, XL generated ROS and caused high-level photoinduced cytotoxicity in vitro via initiation of DNA damage, glutathione depletion, and excessive lipid peroxidation were affective in vivo against HeLa tumor model. Considering our previous results where we applied irradiance doses about 30 J/cm^2^ in in vitro and 90 J/cm^2^ in in vivo experiments—we suppose our formulation may be perspective. Modern IR-triggered PDT drugs applied in clinical practices generally require higher or equal-to XL irradiance doses: Foscan—20 J/cm^2^; Purlytin—200 J/cm^2^; Lutrin, Antrin—25–150 J/cm^2^; Photosens—25–150 J/cm^2^; TOOKAD—5.6–200 J/cm^2^; Verteporfin—50 J/cm^2^; Laserphyrin—up to 260 J/cm^2^. Thus, despite relative long irradiation time (20 min), which is explained by the scheme of applied LED setup, we consider XL and NP–XL formulation as promising PDT agents.

## 4. Conclusions

The aim of this study was to provide a design and characterization rationale for XL encapsulation into PLGA nanoparticles. We demonstrated that specific amounts of XL bind to PVA-enhancing XL photophysical activity; we chose PVA as stabilizer for NP–XL formulation. After the successful scaling of NP–XL formulation, the size of the nanoparticles along with ζ-potential, DL, and EE were suitable for efficient drug delivery. Colorimetric assay, FTIR, DSC, and XRD results confirmed the lack of chemical interactions in NP–XL after encapsulation and the presence of residual PVA after centrifugation with a concentration of 11% (*w*/*w*). After encapsulation, XL exhibited higher photophysical activity compared with XL substance, due to residual PVA influence. NP–XL remained stable during six months of storage, while XL degraded partially upon gamma irradiation. The sterilization via gamma irradiation was unsuitable for NP–XL, so we successfully applied filtration to sterilize the samples. XL and NP–XL exhibited no reactivation of epigenetically repressed genes, indicating NP–XL lacked epigenetic properties. After encapsulation and sterilization, XL retained cytotoxicity against both 2D and 3D tumor cell models indicating the potential of designed formulation in PDT application. Previously, we confirmed the high phototoxicity of NP–XL against MCF-7, HeLa, SKOV-3, A549, and 4T1 cell lines; PS activity retaining by PLGA entrapment; inhibition of colony formation; and induction DNA damage. The main NP–XL cell damaging mechanisms included high levels of ROS generation; apoptosis induction; GSH depletion; lipid peroxidation; and mitochondrial membrane depolarization. The data displayed an activation of ferroptosis and apoptosis pathways after NP–XL-based PDT. Thus, the physico-chemical rationale of XL entrapment into PLGA nanoparticles with high biological efficacy and better applicability of nanoformualtion confirm promise in further studies and applications of NP–XL.

## Figures and Tables

**Figure 1 pharmaceutics-16-00126-f001:**
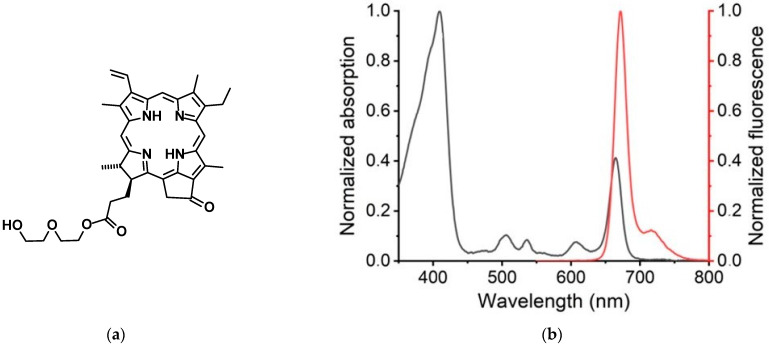
(**a**) Chemical structure of pyropheophorbide α 17-diethylene glycol ester (XL). (**b**) Normalized absorption and fluorescence spectra (λ_ex_ = 510 nm) of XL in acetonitrile. (**c**) XL (1.0 × 10^−6^ M) fluorescence decay in acetonitrile (λ_reg_ = 670 nm). (**d**) Differential triplet-triplet absorption spectrum of XL (3.0 × 10^−7^ M) in degassed acetonitrile (200 μs after flash). Inset depicts XL triplet state kinetic trace at 480 nm (*k*_T_ = 2.5 × 10^3^ s^−1^).

**Figure 2 pharmaceutics-16-00126-f002:**
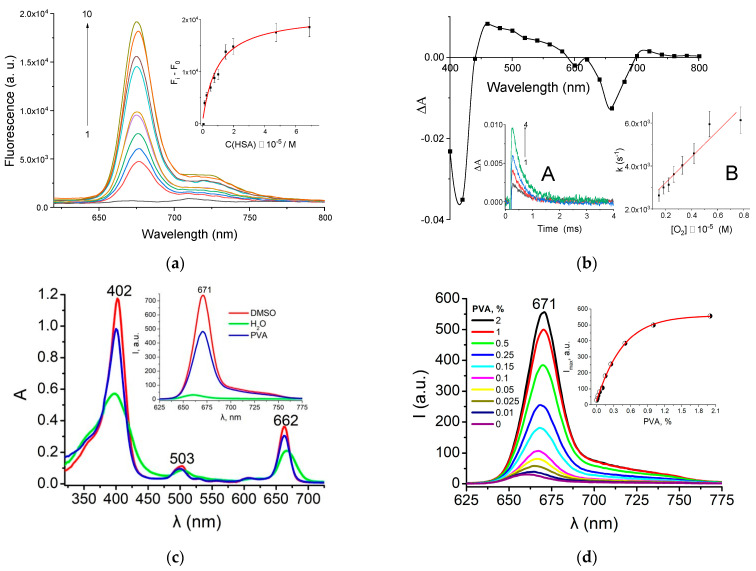
(**a**) XL dye (5 × 10^−7^ M) fluorescence spectra in the presence of different HSA concentration (0–7 × 10^−5^ M) in 0.001 M phosphate buffer pH 7.4. Excitation 410 nm with Xenon lamp. K_b_ = 2.0 × 10^5^ M^−1^. Inset: The XL dye fluorescence intensity (at 670 nm) as a function of HSA concentration. (**b**) Differential absorption spectrum of XL dye (3 × 10^−7^ M) in the presence of HSA (2.0 × 10^−5^ M) in phosphate buffer (150 μs after flash). Inset: A—XL triplet state decay at 460 nm with different HSA (0–2.0 × 10^−5^ M) concentration. B—dependence between triplet decay rate constant and oxygen concentration. (**c**) UV-Vis absorption and fluorescence (insert) spectra of XL (*C* = 6.65 μM) in DMSO, H_2_O and 1% aqueous PVA solution. (**d**) Fluorescence spectra of XL (*C* = 6.65 μM) at different PVA concentrations and XL fluorescence intensity dependence on PVA concentration (insert).

**Figure 3 pharmaceutics-16-00126-f003:**
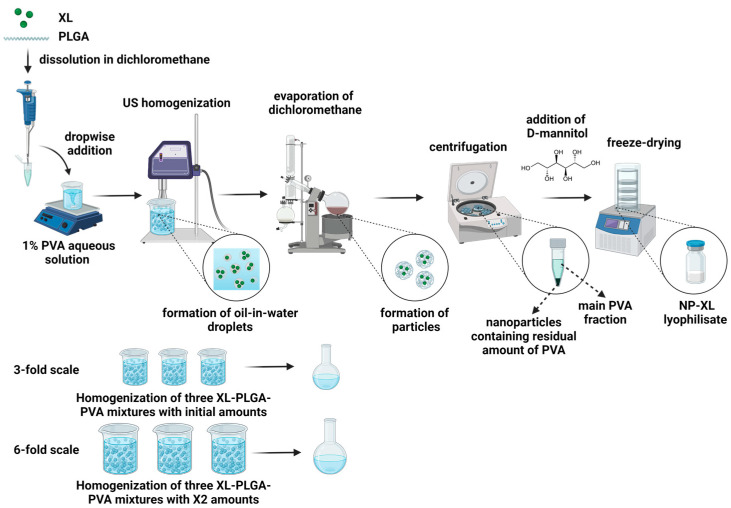
Scheme of synthesis and scale-up of NP–XL (created with BioRender https://www.biorender.com/ (accessed on 18 April 2023)).

**Figure 4 pharmaceutics-16-00126-f004:**
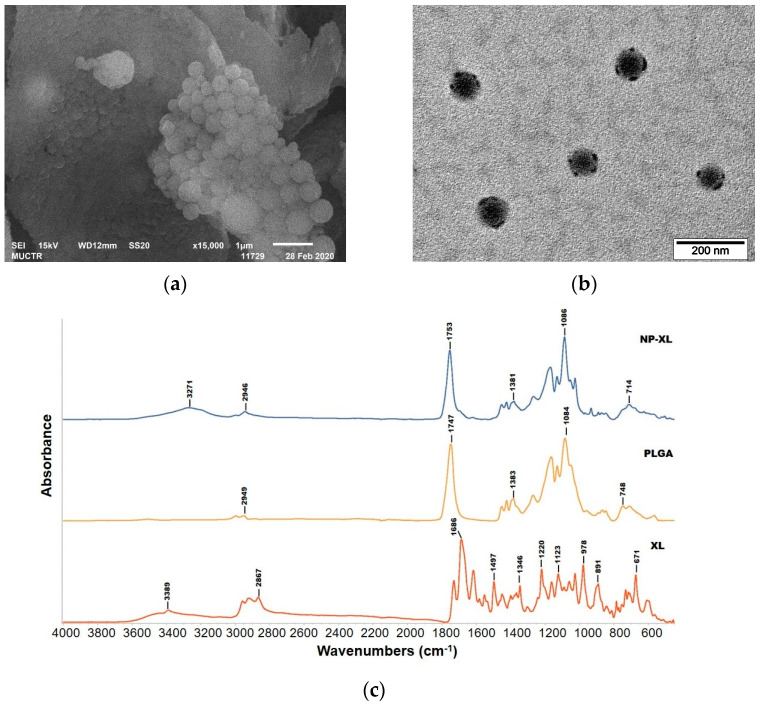
(**a**) SEM with the scale bar of 1 µm and (**b**) TEM with the scale bar of 200 nm of NP–XL. (**c**) FTIR spectra of XL, PLGA, and NP–XL. (**d**) DSC thermograms of PLGA (1), D-mannitol (2), XL (3), XL–NP (4), and PVA (5) (**c**). (**e**) HPLC chromatogram of XL (red) and NP–XL (green).

**Figure 5 pharmaceutics-16-00126-f005:**
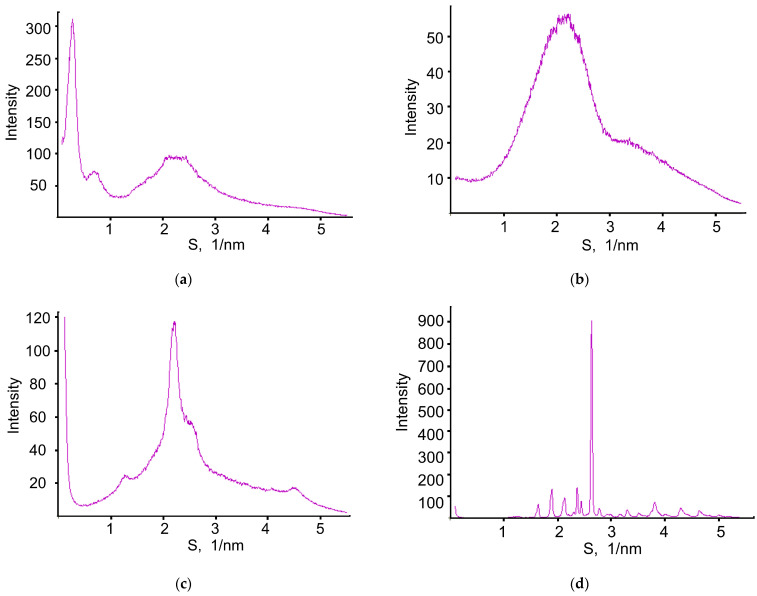
XRD spectra of (**a**) XL, (**b**) PLGA, (**c**) PVA, (**d**) D-mannitol, (**e**) NP–XL.

**Figure 6 pharmaceutics-16-00126-f006:**
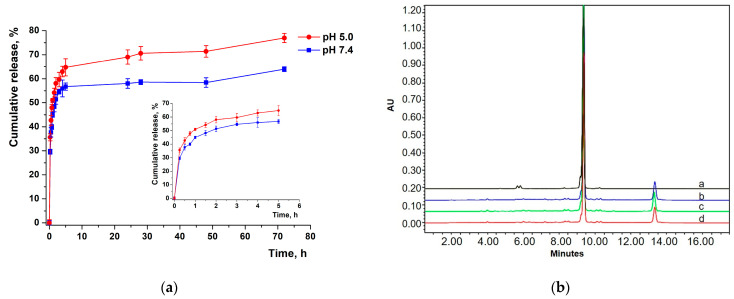
(**a**) In vitro release profile of XL from NPs in 0.01 M PBS pH 7.4 and 5.0 with 0.1% Tween 80. Each point shows mean ± SD (n = 3). (**b**) HPLC chromatograms of (a) NP–XL, (b) NP–XL after irradiation at a dose of 10 kGy, (c) NP–XL after irradiation at a dose of 15 kGy, (d) NP–XL after irradiation at a dose of 25 kGy.

**Figure 7 pharmaceutics-16-00126-f007:**
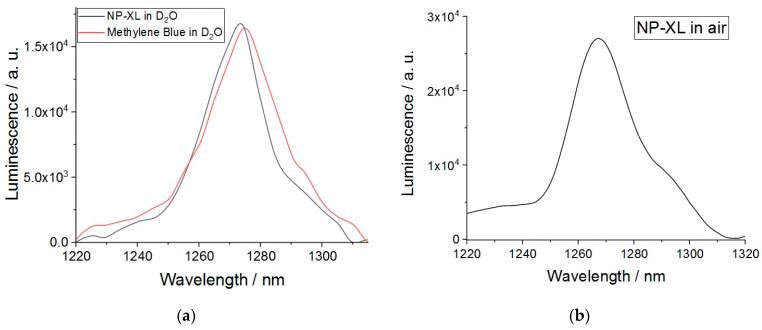
(**a**) Singlet oxygen luminescence in NP–XL solution in D_2_O (λ_ex_ = 675 nm). (**b**) Singlet oxygen luminescence of NP–XL (λ_ex_ = 410 nm).

**Figure 8 pharmaceutics-16-00126-f008:**
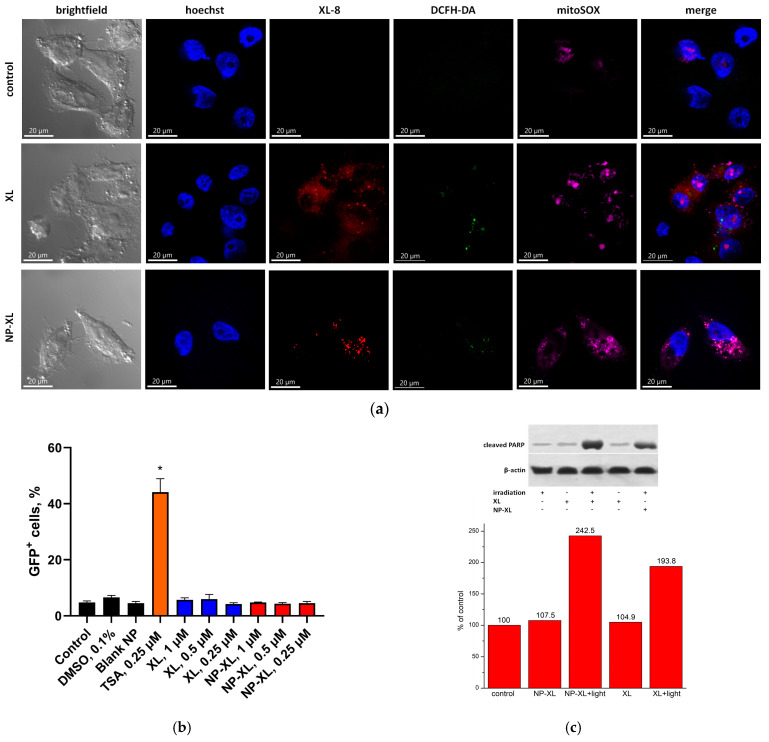
(**a**) Fluorescent images of MDA-MB-231 cells. Subcellular localization analysis was performed after treatment with XL (100 nM) and NP–XL (100 nM). Fluorescent probes–mitoSOX, DCFH-DA, and Hoechst 33342. (**b**) Reactivation of the *GFP* gene in HeLa TI cells analyzed by FACS. Percent of GFP+ cells were determined relative to vehicle control. TSA—positive control. * significant difference in the percentage of GFP+ cells after treatment compared to the negative control. (**c**) Western blot analysis of marker of apoptosis (cleaved PARP) in irradiated or non-irradiated MDA-MB-231 cells treated with XL (100 nM) or NP–XL (100 nM) (the cells were harvested 3 h after treatment).

**Figure 9 pharmaceutics-16-00126-f009:**
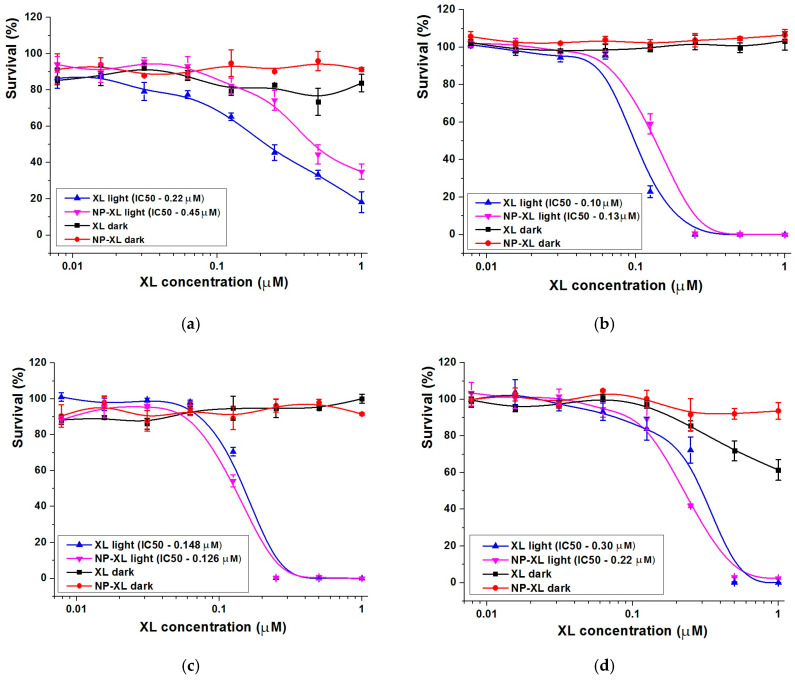
Cytotoxicity of XL and NP–XLs analyzed by MTT assay. The cell viability was evaluated relative to untreated control. MDA-MB-231 (3D) (**a**), MDA-MB-231 (2D) (**b**), K-562 (2D) (**c**), and BJ-5ta (2D) (**d**) measured 72 h after treatment.

**Figure 10 pharmaceutics-16-00126-f010:**
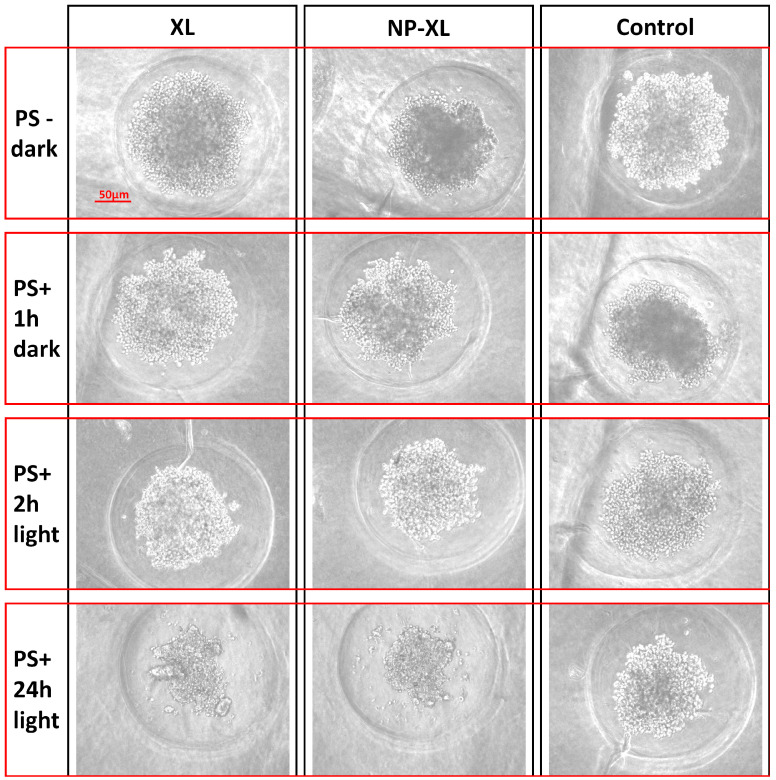
Photo of MDA-MB-231 3D culture (spheroids) with XL (**left column**) and NP–XL (**middle column**) analyzed by MTT assay. The cell viability was evaluated relative to untreated control (**right column**). The images were taken 24 h after cells seeding into plate and just before media supplementation with XL and NP–XL (1st row—PS-dark) (dark); 1 h after media supplementation with XL and NP–XL (2nd row—PS+ 1 h dark) (dark); just 2 h after incubation in presence XL and NP–XL followed by 20 min light-irradiation (3rd row—PS+ 2 h light); 24 h after 2 h incubation in presence XL; and NP–XL followed by 20 min light-irradiation (4th row—PS+ 24 h light). Control was subjected only to light irradiation (**right column**). The survival was evaluated relative untreated MDA-MB-231 spheroids. Bar 50 µm.

**Table 1 pharmaceutics-16-00126-t001:** The influence of batch size on NP–XL parameters. Data shown as mean ± SD (n = 3).

Lab Scale	Size, nm	PDI	ζ-Potential, mV	DL, %	EE, %
1-folds	182 ± 19	0.129 ± 0.010	−22.2 ± 3.8	4.6 ± 0.2	88 ± 3
**3-folds**	**222 ± 10**	**0.243 ± 0.012**	**−12.8 ± 1.4**	**6.6 ± 0.3**	**77 ± 2**
6-folds	295 ± 7	0.310 ± 0.010	−12.8 ± 2.6	5.3 ± 0.3	63 ± 4

**Table 2 pharmaceutics-16-00126-t002:** Results of PVA residual content determination. Data shown as mean ± SD (n = 3).

Initial PVA Concentration, % *w*/*w*	Blank NPs without Centrifugation, % *w*/*w*	Blank NPs after Centrifugation, % *w*/*w*	NP–XL after Centrifugation, % *w*/*w*
88.8 ± 0.3	63.9 ± 0.5	9.4 ± 0.4 *	10.9 ± 0.5 *

* *p* < 0.05 compared to blank NPs without centrifugation.

**Table 3 pharmaceutics-16-00126-t003:** Model fitting of in vitro XL release kinetics.

	Mathematical Model										
	Zero Order	First Order	Higuchi	Hixson–Crowell	Korsmeyer–Peppas			Weibull			
	*R* ^2^	*R* ^2^	*R* ^2^	*R* ^2^	*R* ^2^	*n*	*K*	*R* ^2^	*α*	*β*	*Ti*
NP–XL	−9.4276	−3.9656	−4.9590	−7.3557	0.9905	0.089	44.222	0.6754	1.594	0.115	0.244

**Table 4 pharmaceutics-16-00126-t004:** NP–XL parameters after sterilization. Data shown as mean ± SD (n = 3).

γ-Irradiation Dose, kGy	Size, nm	PDI	ζ-Potential, mV	DL, %	EE, %
Before irradiation	222 ± 10	0.243 ± 0.012	−12.8 ± 4	6.6 ± 0.3	77 ± 2
10	259 ± 92; 2836 ± 1182 *	0.345 ± 0.024 *	−12.0 ± 3	5.4 ± 0.3 *	66 ± 2
15	255 ± 114; 1567 ± 662; 4470 ± 880 *	0.484 ± 0.020 *	−12.4 ± 3	5.1 ± 0.4 *	65 ± 3
25	334 ± 207; 2642 ± 1315 *	0.490 ± 0.031 *	−11.5 ± 3	4.8 ± 0.4 *	65 ± 1

* *p* < 0.05 compared to samples before irradiation.

## Data Availability

Data available on request.

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
