# Peer review of "Pharmaceutical Approach to Develop Novel Photosensitizer Nanoformulation: An Example of Design and Characterization Rationale of Chlorophyll α Derivative"

_pharmaceutics, 2024, doi:10.3390/pharmaceutics16010126_

Round 1

Reviewer 1 Report (Previous Reviewer 1)

Comments and Suggestions for Authors In their manuscript, "Pharmaceutical Approach to Develop Novel Photosensitizer Nanoformulation: An Example of Design and Characterization Rationale of Chlorophyll a Derivative”, Sokol et al. provide a highly detailed account of a nanoformulation containing pyropheophorbide, a photosensitizer derived from chlorophyll a, whose molecule and derivatives are known for their intriguing photodynamic activity. Several modifications were made in the introduction to better contextualize the reader about the work carried out by the group and to further explain the objective of the study presented in this manuscript. The prepared nanoformulation was extensively characterized using various methodologies, showcasing to the reader several interesting properties from a photophysical and photochemical perspective. However, concerning the biological assays, I contend that the conducted tests do not adequately reflect its effective utilization as a potential photosensitizing agent. MTT assays indicate no significant differences in the IC50 values between the unformulated and formulated pyropheophorbide. Therefore, can encapsulation be deemed advantageous? It appears that, biologically, it may not have been beneficial. Exploiting the fluorescence properties of the encapsulated compound, it would be intriguing to understand the intracellular region where the nanoformulation acts. Encouraging confocal microscopy studies using specific organelle probes, such as mitochondria, could provide valuable insights. Could the nanoformulation of the dye interfere with the production of ROS by this compound? To my knowledge, these molecules act through the production of highly cytotoxic ROS at elevated concentrations, such as singlet oxygen and peroxides. The DCFDA assay is common in the context of the photodynamic application of compounds with prominent ROS production. I believe that a comparison with and without formulation is crucial, as it can determine whether encapsulation impacts this property. Based on the literature, it should be noted what type of reactions this nanoformulation produces (whether type I, II, or III). DNA damage is the primary event during photodynamic therapy-induced apoptosis. Since PDT is applied, confirming apoptosis becomes essential. The hallmark of apoptosis remains as PARP cleavage. Determining PARP cleavage through western blot analysis could be more appropriate to confirm PDT-induced apoptosis. Comments on the Quality of English Language

Numerous errors are present throughout the manuscript, encompassing existing content and the newly added text. It is crucial to note that expressions such as "it's" should be written in their expanded form, such as "it is" within the manuscript, adhering to proper grammatical conventions.

Author Response

Dear reviewer,

Thank you for your time and thorough manuscript revision. Below, we have sectioned your comments on the separate questions and  included the answers.

King regards,

The authors

In their manuscript, "Pharmaceutical Approach to Develop Novel Photosensitizer Nanoformulation: An Example of Design and Characterization Rationale of Chlorophyll a Derivative”, Sokol et al. provide a highly detailed account of a nanoformulation containing pyropheophorbide, a photosensitizer derived from chlorophyll a, whose molecule and derivatives are known for their intriguing photodynamic activity. Several modifications were made in the introduction to better contextualize the reader about the work carried out by the group and to further explain the objective of the study presented in this manuscript. The prepared nanoformulation was extensively characterized using various methodologies, showcasing to the reader several interesting properties from a photophysical and photochemical perspective.

1. However, concerning the biological assays, I contend that the conducted tests do not adequately reflect its effective utilization as a potential photosensitizing agent. MTT assays indicate no significant differences in the IC50 values between the unformulated and formulated pyropheophorbide. Therefore, can encapsulation be deemed advantageous? It appears that, biologically, it may not have been beneficial.

Response:

Thank you for your comment. Yes, there was no significant difference between samples according MTT test. One of the main goals of the research was the confirmation that the substance retain activity after nano-formulation process, which was confirmed during our study.

Indeed, there are various FDA and EMA-approved formulations (polymeric and protein particles, liposomes, etc.) available on the market with improved pharmacokinetic and pharmacodynamic properties, which are observable only in vivo – during preclinical or clinical studies.

In vivo research is task of further studies.

2. Exploiting the fluorescence properties of the encapsulated compound, it would be intriguing to understand the intracellular region where the nanoformulation acts. Encouraging confocal microscopy studies using specific organelle probes, such as mitochondria, could provide valuable insights. Could the nanoformulation of the dye interfere with the production of ROS by this compound? To my knowledge, these molecules act through the production of highly cytotoxic ROS at elevated concentrations, such as singlet oxygen and peroxides. The DCFDA assay is common in the context of the photodynamic application of compounds with prominent ROS production. I believe that a comparison with and without formulation is crucial, as it can determine whether encapsulation impacts this property.

Response:

Thank you for your comment. We performed CLSM study and supplemented the manuscript with the images containing information about the substance and formulation subcellular localization in presence of  DCF-DA and mitoSox probes (Figure 8a). We provided concise results discussion concerning the new materials after Figure 8.

3. Based on the literature, it should be noted what type of reactions this nanoformulation produces (whether type I, II, or III).

Response:

Thank you for your comment. The most probable mechanism of action of NP-XL (as well as XL) observed in vitro is the gradual release of XL inside cells. In the free state, the photosensitizer molecule acts mainly as a type 2 photosensitizer [72] – under the influence of light irradiation XL transforms into a triplet state, which was confirmed by flash photolysis experiments. Next, energy is transferred from the triplet state to the oxygen molecule leading to the formation of an active form of oxygen (singlet oxygen).

We supplemented the manuscript with this new information after Figure 8.

4. DNA damage is the primary event during photodynamic therapy-induced apoptosis. Since PDT is applied, confirming apoptosis becomes essential. The hallmark of apoptosis remains as PARP cleavage. Determining PARP cleavage through western blot analysis could be more appropriate to confirm PDT-induced apoptosis.

Response:

Thank you for your comment. We performed WB study, supplemented the manuscript with the image containing information about the PARP cleavage stimulation after the substance and formulation treatment (Figure 8c).

5. English

Numerous errors are present throughout the manuscript, encompassing existing content and the newly added text. It is crucial to note that expressions such as "it's" should be written in their expanded form, such as "it is" within the manuscript, adhering to proper grammatical conventions.

Response:

Thank you for your comment. We corrected the manuscript.

Reviewer 2 Report (New Reviewer)

Comments and Suggestions for Authors

The work has lots of data and solid evidence to support the conclusion basically. The scheme is also detailed and beautiful, however, there’re still some flaws in the manuscript, it’s better to refer to my suggestions to improve the quality before publication.

Major:

1.     In SEM image of NP-XL, the particles were dropped too much, if it’s possible to re-drop with diluted solution and then redo it? In TEM image, there’s some impurities, is it possible to extract more pure NPs and redo the image?

2.     It’s better to supplement the release profile of XL in other pH solutions like pH 6.5 and pH 5.0 since the tumor has a lower pH than normal cells. Maybe it will show a better release effect than pH 7.4.

Minor:

1.     In Figure 2a, it’s better to draw the right axis and label red color to separate the two curves clearly.

2.     In Figure 2a, the x-axis labeling is not correct in the insert figure.

3.     The caption of Fig. 4c is not necessary.

4.     Keep the format uniform, sometimes the authors use “wavelength”, sometimes use “λ”; sometimes “absorption”, sometimes “A”. Please check carefully.

5.     The scheme is nice, it’s better to list which software or tool (I guess it’s Biorender) you use to draw it in the figure caption.

6.     Figures are too many, is it possible to combine some of them together? Like combining figure 6 to figure 9 in one figure. It’s better to keep figures below 10.

Comments on the Quality of English Language

the use of articles has some problems

Author Response

Dear reviewer,

Thank you for your time and thorough manuscript revision. Below, we have sectioned your comments on the separate questions and  included the answers.

King regards,

The authors

The work has lots of data and solid evidence to support the conclusion basically. The scheme is also detailed and beautiful, however, there’re still some flaws in the manuscript, it’s better to refer to my suggestions to improve the quality before publication.

Major:

1. In SEM image of NP-XL, the particles were dropped too much, if it’s possible to re-drop with diluted solution and then redo it? In TEM image, there’s some impurities, is it possible to extract more pure NPs and redo the image?

Response:

Thank you for your comments. We applied filtration via 0.45 µm filter and took the new TEM image. New image is replaced in Fig. 4b. 

In SEM image we tried to find the concentration that provide an analysis of a large number of particles and infer their size and shape in a large sample. While in one part of the image it is difficult to make out individual particles due to gold coating, individual particles in the other part of the image are presented in sufficient quantity and are clearly distinguishable. The image we obtained is similar with previously published images of PLGA nanoparticles, where additional coating with a gold layer was applied [Sharma A. et al. Effects of curcumin-loaded poly (lactic-co-glycolic acid) nanoparticles in MDA-MB231 human breast cancer cells //Nanomedicine. – 2021. – V. 16. – №. 20. – P. 1763-1773; Khan I. et al. Surface Modification and Functionalization of Sorafenib-Loaded PLGA Nanoparticles for Targeting Hepatocellular and Renal Cell Carcinoma. – 2023].

2. It’s better to supplement the release profile of XL in other pH solutions like pH 6.5 and pH 5.0 since the tumor has a lower pH than normal cells. Maybe it will show a better release effect than pH 7.4.

Response:

Thank you for your comments. We performed aditional experiments and supplemented the manuscript with the data (Figure 6a).

Minor:

3. In Figure 2a, it’s better to draw the right axis and label red color to separate the two curves clearly.

Response:

Thank you for your comment. We corrected Figure 2a.

4. In Figure 2a, the x-axis labeling is not correct in the insert figure.

Response:

Thank you for your comment. We supposed you meant the axis label of Figure 2c, so we corrected it.

5. The caption of Fig. 4c is not necessary.

Response:

Thank you for your comment. We corrected the Figure 4 caption (currently Figure 2c, d).

6. Keep the format uniform, sometimes the authors use “wavelength”, sometimes use “λ”; sometimes “absorption”, sometimes “A”. Please check carefully.

Response:

Thank you for your comment. We corrected the manuscript – we replaced “wavelength” with λ in most places, while left “absorption” in the text and “A” in graphs and formulas where it is acceptable.

7. The scheme is nice, it’s better to list which software or tool (I guess it’s Biorender) you use to draw it in the figure caption.

Response:

Thank you for your comment. Yes, we applied Biorender – we added corresponding link in the figure caption.

8. Figures are too many, is it possible to combine some of them together? Like combining figure 6 to figure 9 in one figure. It’s better to keep figures below 10.

Response:

Thank you for your comment. We combined some figures and reduced total number of images down to 10.

Reviewer 3 Report (New Reviewer)

Comments and Suggestions for Authors

In this work, Sokol et al. employed physicochemical properties of novel nanofomulation of photosensitizer – pyropheophorbide α 17-diethylene glycol ester. The main question addressed by the research is how this new photosensitizer can behave in photodynamic therapy. Photodynamic therapy are significant and important topic in clinical practice, so the work fits the current trends in medicine well. The work presents very interesting studies of novel chlorophyll a derivative and its characterisation. The paper is also well written. The text is clear and easy to read.

The studies have been done carefully and no obvious errors or omissions could be detected by referee. The conclusions are consistent with the evidence and arguments presented in materials and methods and then in results and discussion and they addressed the main question posed by the Authors.

My opinion is that the topic is of interest to the readership of Pharmaceutics so the manuscript should be accepted as it is.

Author Response

Dear reviewer,

Thank you for your time and thorough manuscript revision.

King regards,

The authors

Reviewer 4 Report (New Reviewer)

Comments and Suggestions for Authors

Dear authors,

The manuscript presents an approach to develop novel photosensitizer nanoparticles with potential for photodynamic cancer therapy. All methods are described in detail. The results are clearly presented as well and support logically the discussion and conclusions.

To my opinion the article could be accepted in present form after correcting the following remarks. 

1) Figure 4-captions – “(c) Fluores-486 cence spectra of XL in 1% aqueous PVA solution at different XL concentrations”. There is not fig. 4c – needs correction

2) line 711 Isn’t p<0.05?

3) line 713  - Table 6 – seems to be table 4

4) lines 968, 992 – the authors names should be written correctly

5) The authors have to check manuscript for typing errors – lines 123, 458, 532, 897

Kind regards,

the Reviewer

Author Response

Dear reviewer,

Thank you for your time and thorough manuscript revision. Below, we have sectioned your comments on the separate questions and  included the answers.

King regards,

The authors

The manuscript presents an approach to develop novel photosensitizer nanoparticles with potential for photodynamic cancer therapy. All methods are described in detail. The results are clearly presented as well and support logically the discussion and conclusions.

To my opinion the article could be accepted in present form after correcting the following remarks.

1. Figure 4-captions – “(c) Fluores-486 cence spectra of XL in 1% aqueous PVA solution at different XL concentrations”. There is not fig. 4c – needs correction.

Response:

We corrected Figure 4 caption (currently Figure 2 c, d).

2. line 711 Isn’t p<0.05?

Response:

We changed p>0.05 to p<0.05 – indicator of significant difference between groups.

3. line 713 - Table 6 – seems to be table 4

Response:

We corrected the results description after table 4.

4. lines 968, 992 – the authors names should be written correctly

Response:

We corrected the authors names.

5. The authors have to check manuscript for typing errors – lines 123, 458, 532, 897

Response:

We checked and corrected the mentioned issues.

Round 2

Reviewer 1 Report (Previous Reviewer 1)

Comments and Suggestions for Authors

All of my inquiries have been addressed and satisfactorily answered. The inclusion of images using MitoSOX and DCFDA greatly enhanced the manuscript. Noteworthy results were achieved in the PARP cleavage assays. The findings were thoroughly analyzed and interpreted.

Given the successful review of the manuscript, I believe it is now ready for publication.

This manuscript is a resubmission of an earlier submission. The following is a list of the peer review reports and author responses from that submission.

Round 1

Reviewer 1 Report

Comments and Suggestions for Authors

The manuscript by Sokol et al. presents an extensive characterization of a new nanoformulation containing a photosensitizer. This is an already-known molecule and is not recognized as new in this field. The manuscript's most robust characteristics are that it is well written, without significant spelling and typing errors, and, as I mentioned, has a strong component of characterization of the new nanoformulation.

1) The authors state in the introduction that "XL exhibited enhanced hydrophilicity and amphiphilic properties, enabling its integration into various supramolecular assemblies." However, in the next paragraph, they say precisely the opposite: "Despite macrocycle modification, XL still exhibits high lipophilicity, limiting its bioavailability.". A compound cannot be hydrophilic, amphiphilic and lipophilic simultaneously. The authors must be unsure of the properties inherent to this compound's free form. However, structurally, it is highly lipophilic. The authors should clear up this colossal confusion...

2) Regarding cell maintenance, why use the antibiotic gentamicin? This is not what is described as recommended in the manipulation of these cell lines.

3) The introduction focuses heavily on photodynamic therapy, but the paper could be better regarding its PDT application. Although the epigenetic approach in HeLa cells is attractive, only some cellular tests have been carried out, not effectively demonstrating the potential applicability of the nanoformulation that the authors spent so much work characterizing. Since the manuscript focuses on PDT, more than MTT experiments are needed to publish a possible new therapeutic approach.

Therefore, considering mainly this last aspect, and taking into account that the journal Pharmaceutics has a considerable impact factor and is Q1 in the areas of Pharmacology and Pharmaceutical Sciences, it is my opinion to reject the manuscript for publication.

I suggest, within MDPI journals, the transfer of the manuscript to another journal whose purpose is more dedicated to the characterization and creation of new nanoformulations, as well as a slight reformulation of the manuscript or execution of new experimental methodologies that complement the excellent results obtained so far. With the execution of more experimental work using cell lines that show the promise of this nanoformulation, I will agree with the publication in Pharmaceutics.

Author Response

We thank the reviewers for their attention to our study. We have tried to answer all questions as thoroughly as possible and have made appropriate edits to the text of the article. We would like to acknowledge the painstaking work of the reviewers and thank them for their recommendations, as it made the text of the article more structured and competent.

1) The authors state in the introduction that "XL exhibited enhanced hydrophilicity and amphiphilic properties, enabling its integration into various supramolecular assemblies." However, in the next paragraph, they say precisely the opposite: "Despite macrocycle modification, XL still exhibits high lipophilicity, limiting its bioavailability.". A compound cannot be hydrophilic, amphiphilic and lipophilic simultaneously. The authors must be unsure of the properties inherent to this compound's free form. However, structurally, it is highly lipophilic. The authors should clear up this colossal confusion...

We agree with the reviewer and reworded: «Consisting of the hydrophobic moiety and hydrophilic substituent, the structure of XL exhibited amphiphilic properties, which enable its integration into various supramolecular assemblies» and «Despite macrocycle modification, XL still exhibits water solubility limitations, reducing its bioavailability».

2) Regarding cell maintenance, why use the antibiotic gentamicin? This is not what is described as recommended in the manipulation of these cell lines.

Thank you for your valuable comment.

The question of this nature is difficult and seems confusing. The problem is in many reports published by good scientist that are contradictory. Some studies were done in antibiotic free culture media, while others were done in culture media containing the antibiotics added by manufacturer or researchers. There are many factors: different cells lines are characterized with different susceptibility to various antibiotics (e.g. MCF-7 and MDA-MB-231 are less susceptible to gentamycin than MCF-12A [doi: 10.1371/journal.pone.0214586]); antibiotics concentration (many reports describing antibiotics intervention into cellular metabolism study the concentrations much higher than supporting concentrations applied in cell culture [https://doi.org/10.3390/ijms16022307]); primary cell cultures and cultures after transformation generally require antibiotics presence. The presence of many researchers who have access to cell culture facility and presence of low experienced trainees also influences application of antibiotics – the bigger lab, the higher contamination probability. Thus, each laboratory take responsibility individually depending of circumstances and take care of running corresponding controls (in our case all samples were cultured in presence of supporting concentration of gentamycin if not mentioned other).

3) The introduction focuses heavily on photodynamic therapy, but the paper could be better regarding its PDT application. Although the epigenetic approach in HeLa cells is attractive, only some cellular tests have been carried out, not effectively demonstrating the potential applicability of the nanoformulation that the authors spent so much work characterizing. Since the manuscript focuses on PDT, more than MTT experiments are needed to publish a possible new therapeutic approach. Therefore, considering mainly this last aspect, and taking into account that the journal Pharmaceutics has a considerable impact factor and is Q1 in the areas of Pharmacology and Pharmaceutical Sciences, it is my opinion to reject the manuscript for publication. I suggest, within MDPI journals, the transfer of the manuscript to another journal whose purpose is more dedicated to the characterization and creation of new nanoformulations, as well as a slight reformulation of the manuscript or execution of new experimental methodologies that complement the excellent results obtained so far. With the execution of more experimental work using cell lines that show the promise of this nanoformulation, I will agree with the publication in Pharmaceutics.

We thank the Reviewer for valuable comments and for attention to our results. In line with the Reviewer’s opinion, we believe that Pharmaceutics is a highly ranked journal and realize that in some cases results on cell models are essential for article publication.

In our article we provide XL-NP cytotoxicity evaluation on three cell lines - MDA-MB-231, K-562 and BJ-5ta. Moreover, in case of MDA-MB-231 cell line we used both 2D and 3D model (spheroids) and performed the results as cell survival curves and photos of spheroids. While spheroids generation is a time-consuming process, we included cytotoxicity evaluation on 3D model as it provides better mimicking the structure and function of tumors. And this study can be regarded as a link between in vitro and in vivo experiments.

Besides cytotoxicity evaluation we performed XL-NP epigenetic activity study on HeLa TI cell line. We have also included scatter plots for the population corresponding to GFP+ cells in the flow cytometry analysis in accordance with the Reviewer 2 recommendation (Figure S2).

Thus, the present paper summarizes the results of cytotoxicity and epigenetic activity evaluation using four cell lines, including spheroid model.

Previously we have studied the specific activity of XL and XL-NP in vitro and in vivo to show the potential of designed NPs (Mollaeva, M.R.; Nikolskaya, E.; Beganovskaya, V.; Sokol, M.; Chirkina, M.; Obydennyi, S.; Belykh, D.; Startseva, O.; Mollaev, M.D.; Yabbarov, N. Oxidative Damage Induced by Phototoxic Pheophorbide a 17-Diethylene Glycol Ester Encapsulated in PLGA Nanoparticles. Antioxidants 2021, 10, 1985, doi:10.3390/antiox10121985). In present article we focused on XL-NP physicochemical and photophysical properties, considering new in vitro results as additional to previously published. Our results both published and described herein indicate that designed nanoparticles have a potential in PDT and can be considered for further preclinical and clinical studies.

We hope that the Reviewer find the described data interesting and suitable for publication in Pharmaceutics. We also assume that the readers will consider the article as a useful approach of pharmaceutical characterization of similar nanoformulations – PLGA nanoparticles loaded with photosensitizers.

Reviewer 2 Report

Comments and Suggestions for Authors

Dear authors, I recently had the opportunity to review your article titled "Pharmaceutical approach to develop novel photosensitizer nanoformulation: an example of design and characterization rationale of chlorophyll a derivative." First and foremost, I find this article on photodynamic therapy (PDT) applications quite intriguing and valuable. 

While the manuscript contains valuable insights and potential contributions to the field, there are several areas that require substantial improvement to meet the standards of publication. I believe that addressing these issues will significantly enhance the quality and impact of the research.

Below are some of the key points that need attention:

-        In the abstract, it's unclear whether the statement 'For the first time we described that XL binding to polyvinyl alcohol (PVA) enhances XL photochemical activity, providing the rationale for PVA application as a stabilizer for nanoformulations' refers to findings from prior studies or if this is a novel discovery made in your research.

-        Line 68, In vitro should be written in italics (e.g., 'in vitro').

-        In line 80, a spelling error has been identified in the phrase 'approvedfor medical.'

-        The introduction of epigenetic agents is not very clear. It would be beneficial to provide more context related to the topic being addressed.

-        Line 94. Further elaboration is required on what is referred to as the 'favorable effect,' which has been previously described in other articles. Additionally, it is important to specify and thoroughly detail the previous studies that are relevant to the findings presented in this manuscript.

-        Line 97, it appears that the authors are conducting the analysis of the physicochemical properties of the XL-NP complexes and XL in the current manuscript. It is crucial for the authors to emphasize the novelty of this study in relation to the existing literature (references 6-19).

-        In section 2.2.7, the ethical committee code is missing.

-        The authors should provide a justification for the extended incubation time with 660nm light. Effective photosensitizers are expected to act within seconds or a few minutes, and it's important to clarify the reasons for the longer incubation period in this context.

-        Section 3.5. The term 'brightfield TEM' is not clear and needs further clarification.

-        In Figure 7, it is advisable to highlight the 'dark core' mentioned in line 523 for better clarity and understanding.

-        Figure 7 legend appears to be incomplete as it lacks information regarding the identity of each image, the significance of the arrows, and the scale measurements.

-        It would be valuable to specify the size differences between DLS and TEM measurements and provide a justification, as asserted by the authors, attributing this variation to the coating.

-        It is advisable to include scatter plots for the population corresponding to GFP+ cells in the flow cytometry analysis.

-        The epigenetic analysis should specify whether it is conducted after irradiation or not. Furthermore, it is recommended to enhance the cytometry analysis by including a confocal microscopy study that illustrates the GFP+ cells and their morphology in comparison to cells that do not express the fluorescent protein.

-        In Section 3.14, the results are not clearly delineated in connection with the presented figures, which makes it challenging to follow the analysis as currently presented.

-        Figure 15 should include information regarding the effects of the compounds in the absence of irradiation (i.e., in the dark) before treatment. Furthermore, it is essential to display the impact of irradiation on cells without treatment as these controls are crucial for making meaningful comparisons with the presented results.

-        Figure 16 appears to lack clarity in terms of the timing of treatments, making it challenging to interpret. I would strongly recommend conducting a more detailed study that includes higher magnification images to depict cellular morphology. This is essential as it can be difficult to observe the treatment's impact on cell morphology at such low magnifications.

-        The authors should consider discussing their results in comparison to similar photosensitizers and those currently employed in clinical practice. This comparative analysis is crucial to underscore the potential advancements that could be realized in clinical applications with the use of these complexes. Furthermore, it is important to address the required irradiation time and the advantages of a 20-minute irradiation compared to other compounds that exhibit more rapid action.

Author Response

We thank the reviewers for their attention to our study. We have tried to answer all questions as thoroughly as possible and have made appropriate edits to the text of the article. We would like to acknowledge the painstaking work of the reviewers and thank them for their recommendations, as it made the text of the article more structured and competent.

-  In the abstract, it's unclear whether the statement 'For the first time we described that XL binding to polyvinyl alcohol (PVA) enhances XL photochemical activity, providing the rationale for PVA application as a stabilizer for nanoformulations' refers to findings from prior studies or if this is a novel discovery made in your research.

This is our novel discovery that we described in current research. We changed «For the first time we described that XL binding…» to «In our research we revealed for the first time that XL binding…».

-  Line 68, In vitro should be written in italics (e.g., 'in vitro').

We added italics: «In vitro experiments showed that XL…»

-  In line 80, a spelling error has been identified in the phrase 'approvedfor medical.'

We corrected the spelling: «…because the polymer is approved for medical application…»

-  The introduction of epigenetic agents is not very clear. It would be beneficial to provide more context related to the topic being addressed.

We have expanded the introduction as recommended by the reviewer.

-  Line 94. Further elaboration is required on what is referred to as the 'favorable effect,' which has been previously described in other articles. Additionally, it is important to specify and thoroughly detail the previous studies that are relevant to the findings presented in this manuscript.

Thank you for your valuable comment. The issue has been addressed – we presented more detailed description of the previous studies (Introduction section).

We added: «Previously, we performed several studies describing formulation and biological effects evaluation of several perspective reactive oxygen species (ROS) modulators based on pheophorbide [6], tetraphenyl-porphyrin metal complexes [13-15], and their polymer entrapped forms. The main problems of these classes of molecules are high lipophilicity, which limits effective application and potential therapeutic effect realization in vitro and in vivo [16]. We demonstrated efficacy of PLGA-entrapment approach, which allowed us: enhance significantly solubility in physiological media; design, develop and validate complex approach to PLGA-nanoparticles containing Me-porphyrins formulation process [13]; stabilize and properly evaluate physico-chemical properties under physiological conditions; perform reproducible and safe analysis of in vitro and in vivo activity of Me-porphyrins [13-15, 17], XL [6] and bacteriopurpurinimide [18]. PLGA nanoparticles containing FeIIICl-tetraphenylporphyrin in presence of ascorbic acid (AA), representing binary catalyst therapy (BCT), system effectively impaired ROS balance in MCF-7, K-562, and MG-63 cancer cells leading to apoptosis in vitro; in vivo, the combination was safe and highly effective against murine P-388 tumor model [15, 17]. BCT system based on MnIII- tetraphenylporphyrin chloride loaded PLGA nanoparticles and AA endocytosed by 4T1, SKOV-3, K-562 and HeLa cancer cells in vitro effectively generating intracellular H2O2 and leading to cell death via caspase 3/9 activation and AIF translocation to nuclei [14]. The photosensitizers based on XL or bacteriopurpurinimide PLGA-loaded nanoparticles revealed high-level photoinduced cytotoxicity in vitro via initiation of DNA damage, glutathione depletion, and excessive lipid peroxidation, and were effective in vivo against HeLa and S-37 tumor models, respectively [6, 18].»

- Line 97, it appears that the authors are conducting the analysis of the physicochemical properties of the XL-NP complexes and XL in the current manuscript. It is crucial for the authors to emphasize the novelty of this study in relation to the existing literature (references 6-19).

Based on published literature we hypothesized that encapsulation of novel photosensitizer XL is a perspective for photodynamic therapy applications. In our previously published study [Mollaeva M. R. et al. Oxidative damage induced by phototoxic pheophorbide a 17-diethylene glycol ester encapsulated in PLGA nanoparticles //Antioxidants. – 2021. – Т. 10. – №. 12. – С. 1985.], we confirmed high XL photoactivity, inducing reactive oxygen species formation, mitochondrial depolarization, and apoptosis. And we showed that XL remain in vitro photoactivity after encapsulation.

Herein we reported for the first time the main physicochemical properties of XL-NP to

  • Provide a rationale on excipient selection;
  • Get a complete understanding of XL-NP’s properties for further prediction of XL-NP’s behavior in the body.

 As highlighted in different aspects of quality by design [Javed M. N. et al. QbD applications for the development of nanopharmaceutical products //Pharmaceutical quality by design. – Academic Press, 2019. – С. 229-253.], these points are crucial in nanopharmaceutical approach since provide a first glimpse on storage stability, sterilization method for parenteral administration, drug dosing and pharmacokinetics.

Thus, in our study we focused on XL-NP further practical application. Listed tests and comprehensive description can be used as specification for nanoformulation quality assurance. This approach would be of interest for similar photosensitizers’ nanoformulation.

We added above mentioned to the text of the article.

-  In section 2.2.7, the ethical committee code is missing.

Thank you for your valuable comment.

We added: «The experiments including human and animal tissues or cell cultures were con-ducted according to the guidelines of the Declaration of Helsinki, and approved by the Institutional Ethics Committee of N.M. Emanuel institute of biochemical physics (pro-tocol code 29 approved 16.02.2023).»

-  The authors should provide a justification for the extended incubation time with 660nm light. Effective photosensitizers are expected to act within seconds or a few minutes, and it's important to clarify the reasons for the longer incubation period in this context.

We have provided the clarification (section 3.14.). In our opinion the experiment design depends on many factors – hardware, photosensitizer molecular structure and environment. In our case, low-power LED plate allowed us to escape excessive heat generation, application of cooler and light filter. Entrapment of the photosensitizer into polymer matrix can influence the activity interfering with cells survival depending of rerelease profile. Presence of light-dispersing particles and thickness of the media layer can influence light delivery efficacy. Thus, each case (LED setup, photosensitizer and media (nanoparticles, films, etc.)) demands individual adjustment. In our laboratory, we performed preliminary studies of the various light-emitting regimes, and applied one that did not influence untreated cells survival and morphology.

We clarified and emphasized this moment in the manuscript (section 3.14.). In our opinion, the individual LED setup light-exposure time adjustment process itself do not represent any interest for the readers.

Indeed, in different reports related to PDT, we observe inconsistency which really depends on the many factors mentioned above:

  • 30 min exposure https://doi.org/10.1016/j.dyepig.2023.111426;
  • 10 min exposure https://doi.org/10.1016/j.pdpdt.2019.101640;
  • 9 min exposure https://doi.org/10.1016/j.pdpdt.2018.07.011;
  • 5 min exposure doi: 10.1016/j.mtbio.2022.100316;
  • 30-60 min exposure 10.3390/molecules28031433;
  • Up-to 26 min exposure https://doi.org/10.1016/j.pdpdt.2017.11.002;
  • Up to 48 h exposure doi: 10.1038/s41598-017-15142-w.

-  Section 3.5. The term 'brightfield TEM' is not clear and needs further clarification.

According to the literature, the term 'bright field TEM' is «the traditional imaging mode for TEM wherein electrons that are transmitted through the sample without much deflection are used to construct the image» [Klein N. D. et al. Dark field transmission electron microscopy as a tool for identifying inorganic nanoparticles in biological matrices //Analytical chemistry. – 2015. – V. 87. – №. 8. – P. 4356-4362]. However, we decided to delete the term «bright-field» to avoid text overloading.

-  In Figure 7, it is advisable to highlight the 'dark core' mentioned in line 523 for better clarity and understanding.

We highlighted the 'dark core' by white arrows in Figure 7b. We added «…high mass density (dark core marked by white arrows) indicating probably XL core location».

-  Figure 7 legend appears to be incomplete as it lacks information regarding the identity of each image, the significance of the arrows, and the scale measurements.

 We provided the identity of each image as indicated in instructions for authors: SEM image is indicated as a) and TEM image is indicated as b).

Regarding significance of the arrows we used color yellow to describe whole particle size and color white to describe the size of dark core of the particles. We added: «TEM image of NP-XL revealed areas with high mass density (dark core marked by white arrows) indicating probably XL core location»

To make scale more clear we added; «SEM with the scale of 1 µm (a) and TEM with the scale of 100 nm (b) of NP-XL» to figure 7 capture.

-  It would be valuable to specify the size differences between DLS and TEM measurements and provide a justification, as asserted by the authors, attributing this variation to the coating.

Since the main task of microscopy was morphological evaluation of NPs, we decided to delete the lines on size differences between DLS, TEM and SEM measurements. We provided our short description without specification as additional information and decided to delete it to avoid text overloading.

-  It is advisable to include scatter plots for the population corresponding to GFP+ cells in the flow cytometry analysis.

We have added a corresponding drawing to the Supplementary materials (Figure S2).

-  The epigenetic analysis should specify whether it is conducted after irradiation or not.

Analysis of epigenetic activity was carried out after irradiation. Relevant information has been added to the materials and methods.

- Furthermore, it is recommended to enhance the cytometry analysis by including a confocal microscopy study that illustrates the GFP+ cells and their morphology in comparison to cells that do not express the fluorescent protein.

We thank the reviewer for the comment. In our paper "HeLa TI cell-based assay as a new approach to screen for chemicals able to reactivate the expression of epigenetically silenced genes" (doi.org/10.1371/journal.pone.0252504), HeLa TI cell-based assay was verified to analyze the epigenetic activity of xenobiotics. This article presents microscopic images of HeLa TI cells, that demonstrate morphology of the cells after exposure to epigenetically inactive substances and epigenetic modulators (TSA). In this study, flow cytometry was the preferred method because we can analyze 10,000 cells in each sample, allowing adequate statistical analysis.

-  In Section 3.14, the results are not clearly delineated in connection with the presented figures, which makes it challenging to follow the analysis as currently presented.

We corrected figures 15, 16 and discussion – so, the section looks in more obvious and clear way.

-   Figure 15 should include information regarding the effects of the compounds in the absence of irradiation (i.e., in the dark) before treatment. Furthermore, it is essential to display the impact of irradiation on cells without treatment as these controls are crucial for making meaningful comparisons with the presented results.

We observed lack of influence of the XL and NP-XL on the 2D and 3D cell cultures in the dark – we have not reached IC50 values within whole concertation range (the data present at Figure 15). We corrected Figure 15 – now it should look in more obvious and clear way.

The light emitting regime was selected thus it did not influence cell survival, growth rate and morphology. We added: “The light-irradiation mode was selected previously by the way it did not influence cell morphology and survival (data not shown).”

-   Figure 16 appears to lack clarity in terms of the timing of treatments, making it challenging to interpret. I would strongly recommend conducting a more detailed study that includes higher magnification images to depict cellular morphology. This is essential as it can be difficult to observe the treatment's impact on cell morphology at such low magnifications.

We corrected figure 16 and discussion – so, the section looks in more obvious and clear way.

-   The authors should consider discussing their results in comparison to similar photosensitizers and those currently employed in clinical practice. This comparative analysis is crucial to underscore the potential advancements that could be realized in clinical applications with the use of these complexes. Furthermore, it is important to address the required irradiation time and the advantages of a 20-minute irradiation compared to other compounds that exhibit more rapid action.

We added corresponding discussion at the end of 3.14 section.

Reviewer 3 Report

Comments and Suggestions for Authors

The paper “Pharmaceutical approach to develop novel photosensitizer nanoformulation: an example of design and characterization  rationale of chlorophyll a derivative” submitted to Pharmaceutics presents studies regarding the photophysical and photochemical characterization of a photosensitizer nanoformulation based on pyropheophorbide α 17-diethylene glycol ester encapsulated in PLGA nanoparticles. The synthesized nanocarriers were tested on 2D and 3D tumor cell models and showed good potential for PDT applications.

The manuscript is well written, the studies are clearly and in detail presented, the results are of interest for PDT research.

I recommend that the manuscript is worthwhile to be published with some observations as follows:

-          In the abstract/introduction, it should be mentioned that pyropheophorbide a 17-diethylene glycol ester (XL) is a chlorophyll derivative. Chlorophyll derivative appears only in title and keywords.

-          Row 28, abbreviate PLGA

-          Row 93, it is reference [6] not [13].

-          Row 100, correct “photophysical”

-          Row 161, reformulate “difference absorbance changes”

-          Row 164, specify the company, city, country for the PMT

-          Row 214, PDI, DL, EE appear first time, the entire name should be given.

-          Row 414, use for mks, SI units (and everywhere)

-          Figure 3, give the error bars on the fitting curves.

-          Row 431, there are photophysical properties

-          Row 551, reformulate sentence “The presence of an encapsulated drug lacked significant influence on its separation”

-          Rows 566- 568, Figure 8, how you explain “NP-XL spectrum was similar to that one of PLGA, except the fused peak at 2946 cm-1 (С-Н stretching) and the slightly shifted broad peak at 3271 cm-1 (O-H stretching).”  and that no other features of XL appear in the NP-XL spectrum.

-          Row 574, correct “is a useful tool polymers’ composite analysis”

-          Figure 10, improve the quality of the graphs, axes title, numbers are not so visible

-          Figure 11, what are the differences between the results from here and the ones from paper [6].

-          Row 720, replace luminescence with phosphorescence

-          Row 771, the irradiation dose was the same for 2D and 3D cultures?

-          Figure 15, modify the graphs to have the same colour for one test in all graphs (i.e. XL dark is black in a), blue in b), etc); also keep the same order in the legend for all 4 graphs

-           Fig 16, can you improve the contrast of pictures?

-          Rows 809-810, reformulate the phrase “NP-XL remained stable during six month of storage, while XL degraded partially upon gamma-irradiation, but we successfully applied filtration to sterilize the samples.”

Comments on the Quality of English Language

Some phrases should be reformulated to be clearer.

Author Response

We thank the reviewers for their attention to our study. We have tried to answer all questions as thoroughly as possible and have made appropriate edits to the text of the article. We would like to acknowledge the painstaking work of the reviewers and thank them for their recommendations, as it made the text of the article more structured and competent.

-  In the abstract/introduction, it should be mentioned that pyropheophorbide a 17-diethylene glycol ester (XL) is a chlorophyll derivative. Chlorophyll derivative appears only in title and keywords.

We added « … photosensitizer - pyropheophorbide α 17-diethylene glycol ester (XL), which is chlorophyll α derivative, unrevealing insights … » in the abstract and «… reported the synthesis of a novel chlorophyll α derivative called pyropheophorbide α 17-diethylene glycol ester (XL) … » in the introduction.

-  Row 28, abbreviate PLGA

We added PLGA abbreviation: «Our formulation based on poly(lactic-co-glycolic acid) (PLGA) nanoparticles…»

-  Row 93, it is reference [6] not [13].

We changed reference [13] to [6] : «Previously, we studied XL and XL-NP biological activity [6]»

-  Row 100, correct “photophysical”

We corrected the spelling: «…a detail investigation of XL photophysical properties…»

-  Row 161, reformulate “difference absorbance changes”

We changed «difference absorbance changes« to «absorbance changes»

-  Row 164, specify the company, city, country for the PMT

We added «..PMT (Hamamatsu Photonics, Shizuoka, Japan).»

-  Row 214, PDI, DL, EE appear first time, the entire name should be given.

We added the entire names: «Samples were withdrawn after month 1, 3 and 6; diluted with water and analyzed for particle size, polydispersity index (PDI), ζ-potential, drug loading (DL) and encapsulation efficiency (EE)»

-  Row 414, use for mks, SI units (and everywhere)

We changed mks to μs everywhere.

-   Figure 3, give the error bars on the fitting curves.

We indicated error bars on the fitting curves in Figure 3.

-   Row 431, there are photophysical properties

We changed «photochemical» to «photophysical»: «The investigation of XL-HSA photophysical properties showed…»

-  Row 551, reformulate sentence “The presence of an encapsulated drug lacked significant influence on its separation”

We reformulated the sentence: «The presence of an encapsulated drug lacked significant influence on residual PVA concentration»

-  Rows 566- 568, Figure 8, how you explain “NP-XL spectrum was similar to that one of PLGA, except the fused peak at 2946 cm-1 (С-Н stretching) and the slightly shifted broad peak at 3271 cm-1 (O-H stretching).”  and that no other features of XL appear in the NP-XL spectrum.

When comparing the spectra of NP-XL and XL we supposed that the shift of peaks corresponded to С-Н and O-H groups can be due to physical interactions between XL and the polymer. Shifted broad peak at 3271 cm-1 (O-H stretching) also can be attributed to water molecules presented onto NPs surface. Previous studies revealed that lyophilized nanoparticles still contained 3.5% of residual moisture [Passerini N., Craig D. Q. M. An investigation into the effects of residual water on the glass transition temperature of polylactide microspheres using modulated temperature DSC //Journal of Controlled Release. – 2001. – V. 73. – №. 1. – P. 111-115.], which is acceptable and results in only non-significant PLGA degradation [D’Souza S., Dorati R., DeLuca P. P. Effect of hydration on physicochemical properties of end-capped PLGA //Advances in Biomaterials. – 2014. – V. 2014.]. Other characterization peaks of XL are absent in NP-XL spectrum because stronger signals of PLGA chemical bonds mask FTIR signals of encapsulated XL, since the latter are present on the surface of NPs in less quantity than the polymer.

 We added: «Peak at 3271 cm-1 (O-H stretching) also could be attributed to water molecules presented on NP-XL surface. Previous studies revealed that lyophilized nanoparticles still contained 3.5% of residual moisture [42], which is acceptable and results in only non-significant PLGA degradation [43]. The absence of certain characterization peaks of XL in the NP-XL spectrum is due to the stronger signals of PLGA chemical bonds. These signals mask the FTIR signals of the encapsulated XL, as the latter is present in smaller quantities on the surface of the nanoparticles compared to the polymer»

-  Row 574, correct “is a useful tool polymers’ composite analysis”

We corrected: «DSC is a useful tool polymers’ composite analysis…»

-  Figure 10, improve the quality of the graphs, axes title, numbers are not so visible

We improved quality in Figure 10 graphs, axes title, numbers.

-  Figure 11, what are the differences between the results from here and the ones from paper [6].

The results showed in Figure 11 are the same for paper [6] and present article. We decided to repeat them here to provide a consistent presentation and explanation of the results of mathematical modeling. However, we can delete the Figure 11 if necessary.

-  Row 720, replace luminescence with phosphorescence

We changed «luminescence» to «phosphorescence»: «The phosphorescence of molecular oxygen could be registered»

-  Row 771, the irradiation dose was the same for 2D and 3D cultures?

Yeas, we treated 2D and 3D cultures with the same dose. We added: «The irradiation dose was the same for 2D and 3D cultures»

-  Figure 15, modify the graphs to have the same colour for one test in all graphs (i.e. XL dark is black in a), blue in b), etc); also keep the same order in the legend for all 4 graphs

We modified the graphs in Figure 15.

-  Fig 16, can you improve the contrast of pictures?

We improved the contrast of the pictures in Figure 16.

-  Rows 809-810, reformulate the phrase “NP-XL remained stable during six month of storage, while XL degraded partially upon gamma-irradiation, but we successfully applied filtration to sterilize the samples.”

We reworded this sentence: «NP-XL remained stable during six month of storage, while XL degraded partially upon gamma-irradiation. As sterilization with gamma-irradiation was unsuitable for NP-XL, we successfully applied filtration to sterilize the samples».

Round 2

Reviewer 1 Report

Comments and Suggestions for Authors

Thanks to the authors for the answers. Still, I miss more studies demonstrating how these nanoformulations effectively work after photodynamic treatment: what is the death mechanism, etc...

Is there no possibility of additional tests being carried out?

An extensive characterization of the particles, with just a few cellular tests, does not enhance the therapeutic potential of the formulations.

I suggest transferring the manuscript to the journal Pharmaceuticals if additional tests cannot be carried out.

Reviewer 2 Report

Comments and Suggestions for Authors

1.     The authors do not indicate the specific changes in the new version, i.e., they do not specify the line numbers where they have included the new comments. This is crucial for the reviewer to see the changes and the context in which the authors intend to refer to them. I kindly request the exact incorporation of changes in the new version of the manuscript to facilitate the review process.

2.     In the introduction section, authors do not specify what they have expanded upon. In my opinion, the introduction is now overly lengthy and does not provide the key points I requested in my initial review.

3.     The legend for Figure 7 is still incomplete. The reader may not necessarily understand what is depicted in the image unless the authors provide an explanation.

4.     "HeLa TI cell-based assay as a new approach to screen for chemicals able to reactivate the expression of epigenetically silenced genes" (doi.org/10.1371/journal.pone.0252504). In this article, there is still a lack of confocal microscopy images and detailed cellular morphology as I previously requested in my initial review. I kindly request that a more comprehensive study be conducted, including the incorporation of confocal microscopy to provide a more in-depth analysis of cellular morphology.

5.     The Figure 15 is currently quite confusing. In the legend, the authors mention 'XL light (IC 50-0.22 micrM),' but the X-axis of the graph displays the concentration of XL. It's unclear what the actual concentration used is. Could the authors please clarify this discrepancy and provide the accurate concentration of XL used in the experiment?

6.     The Figure 16 appears to remain largely unchanged; the authors have simply enlarged the images without providing any additional annotations. In fact, this has made the figure even more confusing than before. I kindly request that the authors include higher-resolution images that allow for a more detailed analysis of cellular morphology.